# CAGE: Probing Causal Relationships in Deep Generative Models

## Abstract

Deep generative models excel at generating complex, high-dimensional data, often exhibiting impressive generalization beyond the training distribution. The learning principle for these models is however purely based on statistical objectives and it is unclear to what extent such models have internalized the causal relationships present in the training data, if at all. With increasing real-world deployments, such a causal understanding of generative models is essential for interpreting and controlling their use in high-stake applications that require synthetic data generation. We propose CAGE, a framework for inferring the cause-effect relationships governing deep generative models. CAGE employs careful geometrical manipulations within the latent space of a generative model for generating counterfactuals and estimating unit-level generative causal effects. CAGE does not require any modifications to the training procedure and can be used with any existing pretrained latent variable model. Moreover, the pretraining can be completely unsupervised and does not require any treatment or outcome labels. Empirically, we demonstrate the use of CAGE for: (a) inferring cause-effect relationships within a deep generative model trained on both synthetic and high resolution images, and (b) guiding data augmentations for robust classification where CAGE achieves improvements over current default approaches on image datasets.

## 1 Introduction

Causal models play a fundamental role in analyzing real-world datasets, as many of the questions underlying scientific research require reasoning beyond simply associations (Pearl, 2009; Rubin, 2005). Fundamentally, a causal model characterizes the generative process of the observed data via establishing cause-effect relationships between attributes; for example, a disease *causes* a symptom *effect* (and not vice versa). Consequently, a causal analysis is concerned with data generated and collected in the real world via *natural* processes and biological agents (Glymour et al., 2019).

In contrast, a major series of advancements in machine learning in the last decade concerns deep generative models that are learned via statistical objectives to *artificially* generate high-dimensional data. The widespread use of these models across the artificial intelligence spectrum, from vision (Karras et al., 2020) and language (Brown et al., 2020) to scientific discovery (Sanchez-Lengeling & Aspuru-Guzik, 2018) and decision making (Chen et al., 2021), raises an important question: What are the causal mechanisms governing a deep generative model? With a statistical training objective, it is unclear to what extent (if any) the causal relationships from the observed dataset are retained within a generative model and consequently, the outputs expected from the model under interventions and counterfactuals. In addition to controlling and interpreting these black-box models, such a fine-grained understanding can be paramount for formally characterizing their biases and generalization capabilities prior to high-stake deployments.

**Present Work**. We propose CAGE, a framework for inferring implicit cause-effect relationships in the latent space of deep generative models. Our framework builds off the potential outcomes framework of Neyman-Rubin causal model (Neyman, 1923; Rubin, 1974) and defines a new notion of *generative* average treatment effects (GATE). A fundamental problem of causal inference is that, by construction, we cannot observe the potential outcomes under all treatments (Holland, 1986) i.e., at any given time, any individual can be assigned only one treatment (a.k.a. the factual) but not both. However, when studying treatment effects for deep generative models, CAGE exploits the generative nature of such models to overcome this challenge and explicitly generate the counterfactual. In

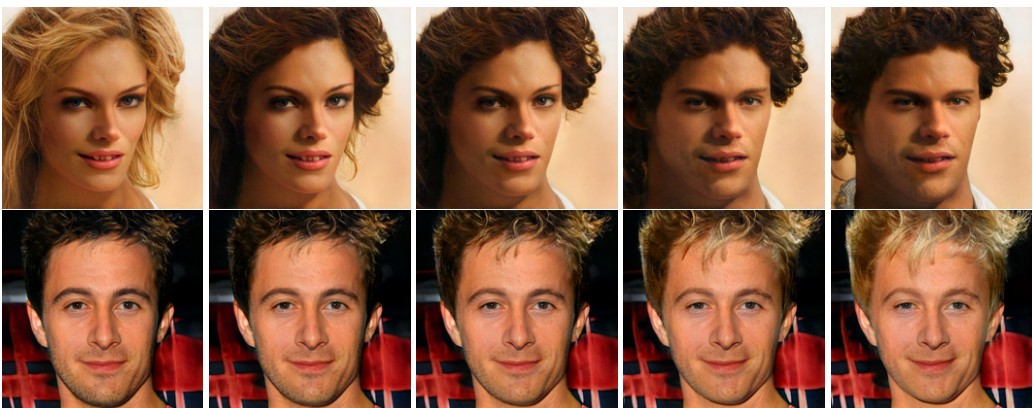

Figure 1: Counterfactual generation of Blonde Males using a deep generative model (Karras et al., 2020). CAGE infers the implicit causal structure as Gender → Blonde Hair. **Top:** The left panel contains the original image of a blond female, other panels display counterfactual samples where gender (treatment) is changed to male. **Bottom:** Equivalent visualizations starting from a non-blond male (left panel) and counterfactually changing hair color (treatment) to blond. Generating counterfactual samples that respect the implicit causal structure yields subjectively better samples.

particular, our approach utilizes probabilistic classifiers for characterizing the axis of variations for relevant attributes in the latent space of a deep generative model. Given the axis for the treatment attribute classifier, we develop a geometric manipulation scheme for simulating the assignment of specific treatments (in particular, the counterfactual) to an individual. Thereafter, we use an outcome attribute classifier to quantify the difference in outcomes for the factual and counterfactual generations and thus, estimate GATE. We refer to our overall framework for causal probing of deep generative models as CAGE. Figure 1 shows illustrative samples generated from our approach.

While there is a body of existing work on enforcing a causal structure on deep generative models e.g., (Kocaoglu et al., 2017; Yang et al., 2021; Suter et al., 2019) or using generative models as tools to aid causal inference e.g., (Louizos et al., 2017; Yoon et al., 2018; Sauer & Geiger, 2021), our work stands out in extracting the causal structure inherent in existing generative models. As such, CAGE applies readily to any pretrained latent variable model, without any changes to their training procedure that can introduce any undesirable trade-offs in learning (Locatello et al., 2019).

We further leverage the treatment effects estimated via CAGE to define a natural measure of causal direction over a pair of variables associated with a deep generative model. Our score is simply defined as the difference in magnitudes of GATE when considering each variable as a *treatment*. The intuition behind such a score is that the ATE when considering the effect as a treatment should be approximately zero. In contrast, we expect the magnitude of the ATE when considering the cause as a treatment to be large. We study the consistency of the inferred causal directions for two high-dimensional image datasets, MorphoMNIST (Pawlowski et al., 2020) and CelebaHQ (Karras et al., 2017), with known or biologically guessed prior cause-effect relationships.

Finally, we use these inferred causal directions to guide counterfactual generation for data augmentations. Data augmentations from generative models are immensely useful for dealing with class and subgroup imbalance (Ramaswamy et al., 2021; Xu et al., 2018; Sattigeri et al., 2018). We show that augmentations generated by respecting the causal directions inferred via CAGE improve classification accuracy by over 2.7% over the baseline and 11.7% over the anti causal direction on the CelebaHQ dataset for the worst subgroup in the challenging group DRO task. Our main contributions are summarized as follows:

1. We introduce the notion of generative average treatment effects (GATE) for probing cause-effect relationships in the latent space of deep generative models.

2. We propose CAGE, a GATE estimation framework for deep generative models based on latent counterfactual manipulations.

3. Finally, as a practical application, we use the ATEs to infer causal direction over pairs of attributes within deep generative models and employ CAGE for generating counterfactual data augmentations with an application to distributionally robust optimization.

## 2 BACKGROUND AND PRELIMINARIES

### 2.1 THE POTENTIAL OUTCOMES FRAMEWORK

We briefly overview the potential outcomes framework of Neyman (1923) and Rubin (1974), upon which we base our proposed method in the next section. In its simplest form, this framework considers the causal effect of assigning a binary treatment, $T \in \{0, 1\}$. Such a framework posits the existence of potential outcomes for the $i$th individual both whilst receiving treatment, $Y_{T=1}(i)$, and when treatment is withheld, $Y_{T=0}(i)$. The causal effect for a given individual, $i$, is defined as the difference in these potential outcomes.

The "fundamental problem of causal inference" is that whilst we may posit the existence of both potential outcomes, $Y_{T=1}(i)$ and $Y_{T=0}(i)$, we only ever observe the outcome under one treatment (Holland, 1986). For this reason, causal inference is often performed at the population level, by considering quantities such as the average treatment effect (ATE):

$$\tau = \mathbb{E}_i \left[ Y_{T=1}(i) - Y_{T=0}(i) \right]. \tag{1}$$

Equation (1) is the expected difference in potential outcomes of individuals receiving treatment $T = 1$ and $T = 0$. Under standard assumptions of unconfoundedness, positivity, and stable unit treatment value assumption, the ATE is identifiable and can be estimated by a statistical estimate of the associational difference in expectations (Rubin, 1978).

### 2.2 DEEP LATENT VARIABLE GENERATIVE MODELS

The goal of generative modeling is to learn an underlying data distribution given a training dataset. For high-dimensional data, this can be done effectively via deep neural networks by learning a mapping $G$ from unobserved factors of variation $\mathcal{Z}$ (i.e., latent variables) to observed variables $\mathcal{X}$ (e.g., raw pixels). The mapping $G$ is parameterized using deep neural networks, and can be learned via a variety of training objectives, such as adversarial training (e.g., generative adversarial networks (GAN; Goodfellow et al. (2014)), exact maximum likelihood estimation (e.g., Normalizing Flows (Dinh et al., 2017)), and variational Bayes (e.g., Variational Autoencoders (VAE; Kingma & Welling (2013); Rezende & Mohamed (2015)). For our current work, we can use any such latent variable generative model. Our experiments will exhibit this generality by considering state-of-the-art flows (Papamakarios et al., 2017; Song et al., 2019) and GANs (Karras et al., 2020).

In addition to $G$ (i.e., the decoder), we will assume the existence of an encoder $\text{ENC} : \mathcal{X} \rightarrow \mathcal{Z}$ that can map any observed datapoint to its latent vector representation. For instance, for flows, the encoder is given by the inverse of the generator mapping, $\text{ENC} = G^{-1}$. For VAEs, the encoder is the variational posterior distribution trained alongside $G$. For GANs, either the encoder can be trained during training (e.g., Donahue et al. (2016); Dumoulin et al. (2016)), after training (e.g., Richardson et al. (2021)), or implicitly learned by backpropagating through $G$ to find a latent vector $z \in \mathcal{Z}$ that best represents a target input $x$, as measured via a reconstruction loss (e.g., Lipton & Tripathi (2017); Bora et al. (2017)) or a pretrained domain classifier (e.g., (Karras et al., 2020)).

## 3 ESTIMATING CAUSAL EFFECTS IN DEEP GENERATIVE MODELS

Our goal is to interrogate a deep generative model with a view to better understanding which—if any—causal associations have been implicitly encoded. We therefore look to answer the question:

*In the context of a generator, $G$, does considering the counterfactual over a given "causal" attribute (e.g., gender) lead to significant changes in a "effect" attribute (e.g., hair color)?*

Formally, we assume whitebox access to a pretrained latent variable deep generative model $G : \mathcal{Z} \rightarrow \mathcal{X}$. In addition, we further assume a finite dataset of $n$ annotated observations, $D = \bigcup_{i=1}^{n} \{\mathbf{x}_D(i), m_1(i), m_2(i)\}$. Here, the examples $\{\mathbf{x}_D(1) \ldots, \mathbf{x}_D(N)\}$ are drawn from the model's training distribution and we annotate each example $\mathbf{x}_D(i)$ with binary metadata for two variables $m_1(i) \in \{0, 1\}$ and $m_2(i) \in \{0, 1\}$. These two annotated variables can be drawn from a larger *unobserved* (w.r.t. $G$) causal process. For example, $\mathbf{x}_D(i)$ could correspond to a high-dimensional image and $m_1(i)$ and $m_2(i)$ could be any two attributes of the image, such as gender expression and hair color in Figure 1. Apriori, we do not know which of the two variables (if any) is cause or effect but through $D$ and $G$, we seek to identify the nature of their relationship within $G$.

We note that any causal conclusions obtained under our proposed framework only reflect implicit properties of a generator $G$, as opposed to properties of the data on which the generator was trained. That is to say, it is perfectly permitted, for $G$ to have learned a causal relationship that is inconsistent with the observational data used for training. As we shall see in §4, shedding light on the implicit causal relationship learned by $G$ is abundantly useful for pinpointing shortcomings in the model and informing best practices for counterfactual generations. As a result, this work—by design— is in sharp contrast with the majority of causal inference literature that focuses on identifying the generative mechanisms of a fixed dataset as opposed to our focus on the generative model itself.

### 3.1 OVERVIEW OF THE CAGE FRAMEWORK

For answering the above question, we propose to extend the potential outcomes framework described in §2.1 to accommodate latent variable deep generative models. Our approach is premised on computing a *generative* average treatment effect for a candidate causal attribute, $m_c$, on a given effect attribute, $m_e$, for a given generator, $G$. Unlike traditional approaches to treatment effect estimation, the key observation in CAGE is that we can simulate the counterfactuals under a given treatment. We refer to our overall framework for causal probing of generative models as CAGE.

For example, in order to investigate the cause-effect relationship between gender and hair color, let us initially arbitrarily assign gender to be the treatment $m_1 = m_c$ (cause) and hair color to be the outcome $m_2 = m_e$ (effect). We note this choice is arbitrary and the overall algorithm will also consider the reverse assignment (i.e., $m_1 = m_e$ and $m_2 = m_c$) and if needed, also reveal independence between the variables. Then we may simulate the effects of treatment assignments by manipulating the gender attribute in the latent space and measuring the corresponding change in outcomes. We next discuss our strategies for counterfactual manipulation and outcome quantification.

### 3.2 SIMULATING TREATMENTS VIA LATENT COUNTERFACTUAL MANIPULATIONS

In order to simulate the assignment of counterfactuals, we follow a three-step procedure.

**Step 1:** Our first step is to train a latent space classifier for the attributes using $D$. Here, we use the encoder to first project every input $\mathbf{x}_D(i)$ in $D$ to its latent encoding $\mathbf{z}_D(i) = \text{ENC}(\mathbf{x}_D(i))$. [1] Given the latent encodings $\mathbf{z}_D(i)$ and their annotations for $m_c$ and $m_e$ attributes, we train a (probabilistic) linear classifier, $\phi_c : \mathcal{Z} \to [0, 1]$, to discriminate between binary causal attribute values given latent representations. Concretely, by restricting ourselves to linear classifiers we obtain a unit hyperplane that encodes the classification boundary. We use $\mathbf{h}_c$ to denote the normal vector to the hyperplane for attribute $m_c$.

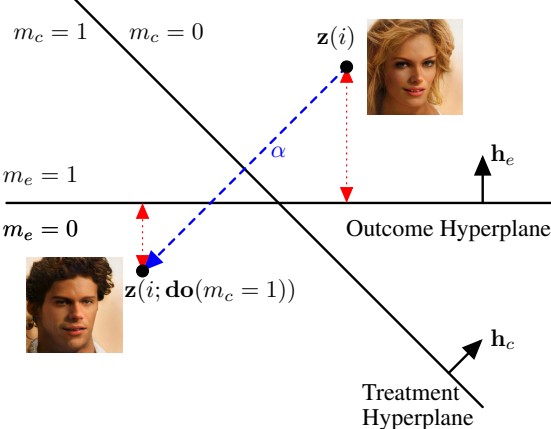

Figure 2: Counterfactual Manipulation in the Latent Space of Generative Models in CAGE.

**Step 2:** Since we are interested in casual discovery for the generative model, in the second step, we use $G$ to create a dataset of annotated examples by sampling $k$ latent vectors $Z = \bigcup_{i=1}^{k} \{\mathbf{z}(i)\}$ from the prior of the generative model. Let $X = \bigcup_{i=1}^{k} \{\mathbf{x}(i) = G(\mathbf{z}(i))\}$ denote the corresponding generations. Since sampling latents from the prior generative model are typically inexpensive, our dataset size can be fairly large. For each generated example $\mathbf{x}(i) \in X$, we obtain its factual treatment by simply using the latent space classifier as $m_c(i) = \mathbb{1}[\phi_c(\mathbf{z}(i)) > 0]$.

**Step 3:** For a given generated sample, $\mathbf{x}(i) \in X$, assume without loss of generality, that $m_c = 0$ is the factual treatment obtained via Step 2. We can derive analogous expressions for factual treatments $m_c = 1$, but we skip those for brevity. Finally, we define the counterfactual latent with respect to setting the treatment attribute $m_c = 1$ as using the **do** operator as:

$$\mathbf{z}(i; \mathbf{do}(m_c = 1)) = \mathbf{z}(i) + \alpha \mathbf{h}_c$$

---

[1] For stochastic encoders such as in VAEs, we consider the mean of the encoding distribution.

where $\alpha \in \mathbb{R}_+$ is a positive scalar that controls the extent to which we manipulate. Equation (3.2) corresponds to moving linearly along the hyperplane normal, $\mathbf{h}_c$, which encodes attribute $m_c$ for simulating the counterfactual. Such an approach is premised on the assumption that $\mathcal{Z}$ is linearly separable with respect to a semantically meaningful latent attribute, which was first observed and empirically validated for GANs (Denton et al., 2019). A counterfactual sample can subsequently be obtained by simply pushing forward the counterfactual latent through $G$:

$$\mathbf{x}(i; \mathbf{do}(m_c = 1)) = G(\mathbf{z}(i; \mathbf{do}(m_c = 1))). \tag{2}$$

Figure 2 shows an illustration of our counterfactual manipulation strategy.

### 3.3 QUANTIFYING TREATMENT EFFECTS VIA PROXY CLASSIFIERS

Equations (3.2-2) define the geometric manipulations required to obtain interventional samples $\mathbf{x}(i; \mathbf{do}(m_c = 1))$ from a generative model, $G$. However, in order to estimate treatment effects, we require an estimate of the presence or absence of the effect attribute, $m_e$, for the counterfactuals. To this end, we propose the use of binary classifiers trained to detect the presence of an effect variable. These classifiers can be trained directly in the latent space and serve to measure the effect of intervening on the treatment attribute $m_c$ for an individual $\mathbf{x}(i)$ be quantified as:

$$m_e(i; \mathbf{do}(m_c = 1)) = \phi_e \left( \mathbf{z}(i; \mathbf{do}(m_c = 1)) \right). \tag{3}$$

Alternatively, we can train another probabilistic classifier $\psi_e : \mathcal{X} \to [0, 1]$ that classifies the counterfactual generations in Equation (2). In this manner, we can then quantify the effect of intervening on the treatment attribute $m_c$ for an individual $\mathbf{x}(i)$ as:

$$m_e(i; \mathbf{do}(m_c = 1)) = \psi_e \left( \mathbf{x}(i; \mathbf{do}(m_c = 1)) \right). \tag{4}$$

For simplicity, we use latent space classifiers for quantifying treatment effects, as in Equation (3). We can define the *generative* individual treatment effect (GITE) estimate $\tau_{\text{GITE}}$ as:

$$\tau_{\text{GITE}}(m_c \to m_e) = m_e(i; \mathbf{do}(m_c = 1)) - m_e(i; \mathbf{do}(m_c = 0)). \tag{5}$$

Taking an expectation over the generated examples, we get an estimate for the *generative* average treatment effect (GATE) of $m_c$ on $m_e$ as:

$$\tau_{\text{GATE}}(m_c \to m_e) = \mathbb{E}_i \left[ m_e(i; \mathbf{do}(m_c = 1)) - m_e(i; \mathbf{do}(m_c = 0)) \right] \tag{6}$$

As we can see from Equation (6), the correctness of our estimate relies on a number of assumptions. In addition to standard assumptions from causality (unconfoundedness, positivity, SUTVA), the quality of the generative model and classifiers play an important role. For the generative model, we designed our counterfactual manipulation scheme assuming that the latent space is linearly separable in the attributes of interest. Further, implicit in our framework is the assumption that the generative model can generalize outside the training set to include the support of the counterfactual distribution. On the use of classifiers, we need accurate and calibrated classifiers (ideally, Bayes optimal) for both manipulating latent vectors and quantifying treatment effects. While it is impossible to test in—full veracity— the above assumptions on real-world distributions, there is significant empirical evidence in the last few years that suggest that modern deep generative models and classifiers can indeed satisfy the above requirements for many practical usecases (Denton et al., 2019; Brown et al., 2020). The theoretical properties and these necessary assumptions of CAGE are further discussed in §A.

$\Delta\tau$ **as a measure of causal direction.** Finally, we note that the preceding sections have focused on the challenge of quantifying if a variable, $m_c$, has a causal effect on second variable, $m_e$. This assumes knowledge of the causal ordering over variables and maybe considered a special case of the more general causal discovery problem. As such, we can extend our approach to define a measure of causal direction over any pair of variables by considering the difference in absolute generative ATE scores when either variable is considered as the treatment:

$$\Delta\tau = | \tau_{\text{GATE}}(m_c \to m_e) | - | \tau_{\text{GATE}}(m_e \to m_c) | \tag{7}$$

The $\Delta\tau$ score determines the magnitude of the difference in outcomes when either attribute is prescribed as the treatment. Intuitively, when using the true causal attribute, according to $G$, as the

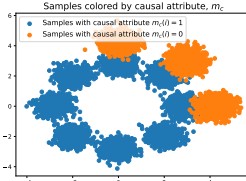 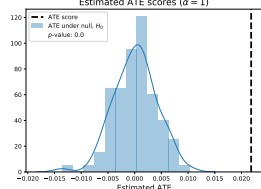 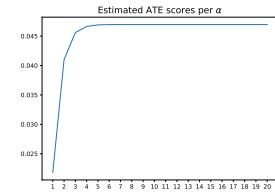

Figure 3: Synthetic example of CAGE applied to a toy dataset. **Left**: A scatterplot with points colored by the value of causal attribute, $m_c$. **Center**: Histogram of GATE values under null hypothesis, $H_0$, and vertical line denoting estimated GATE. **Right**: Plot of $\Delta\tau$ as a function of $\alpha$.

treatment we expect a signature of the interventions to be large in magnitude. Conversely, the magnitude of disturbance measured when manipulating the true effect variable should be markedly smaller. Thus $\Delta\tau > 0$ corroborates that the chosen causal ordering is indeed the correct one as supported by $G$, while a $\Delta\tau < 0$ implies that the chosen causal ordering may in fact be reversed.

**A null distribution for treatment effects.** A further important consideration relates to how we might determine whether an estimated GATE is statistically significant. To address this, we can obtain an empirical sample of GATE scores under the null hypothesis where the attribute $m_c$ has no causal association with $m_e$. A simple manner through which this can be obtained is via randomization and permutation testing. In particular, we can randomly shuffle the values of a causal attribute, $m_c$, thereby removing any potential causal association. This corresponds to performing interventions over the latent space of $G$ which are effectively random projections. This process is repeated many times to obtain an empirical distribution for a GATE under the null hypothesis. Given an empirical distribution of GATEs under the null, we can obtain a $p$-value for our observed GATE.

## 4 EXPERIMENTS

We evaluate the ability of CAGE towards inferring causal relationships in pretrained deep generative models on both synthetic and high-dimensional datasets such as high-resolution images of faces.

### 4.1 SYNTHETIC MIXTURE OF GAUSSIANS

We consider the following toy setup: data is generated according to two distinct mixture distributions as shown in the left panel of Figure 3 where each color denotes a mixture. We define the causal attribute, $m_c$, to be which mixture each sample is drawn from (i.e., from the mixture of 8 Gaussians, in blue, or the mixture of 3 Gaussians, in orange). We further define an effect variable, $m_e$, to be defined as one, if the mean of the $(x, y)$-coordinates is less than 2.5 and zero otherwise. In this manner, we can see that $m_c$ has a causal influence over $m_e$ by determining which mixture each sample is drawn from. We employ a Masked Autoregressive Flow (Papamakarios et al., 2017) as the deep generative model. The middle panel of Figure 3 visualises the distribution of $\tau_{\text{GATE}}$ under the null as well as the estimated GATE, $\tau_{\text{GATE}}(m_c \rightarrow m_e)$, which is significantly larger in magnitude. Finally, the right panel plots $\Delta\tau$ as a function of $\alpha$, positive values corroborate that $m_c \rightarrow m_e$.

### 4.2 IMAGE DATASETS

**Morphomnist**. We evaluate the causal relationships learned by powerful deep normalizing flow models on a synthetic dataset based on MNIST dubbed MorphoMNIST as introduced by Pawlowski

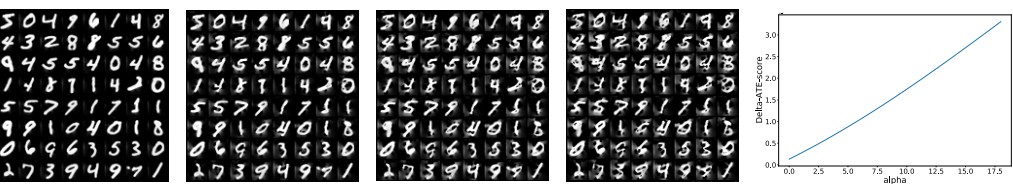

Figure 4: Generating counterfactuals of MorphoMNIST Digits by reducing the thickness of each digit without changing the average intensity. **Right**: Plot of $\Delta\tau$ as a function of $\alpha$.

Table 1: Causal Discovery over $G$ for various pairs of attributes, $(m_1, m_2)$. For each pair, the top row corresponds to taking $m_1$ as the treatment whilst the bottom utilizes $m_2$ as the treatment. Please see appendix §D contains additional generated samples and a qualitative description of visual fidelity for each pair of attributes.

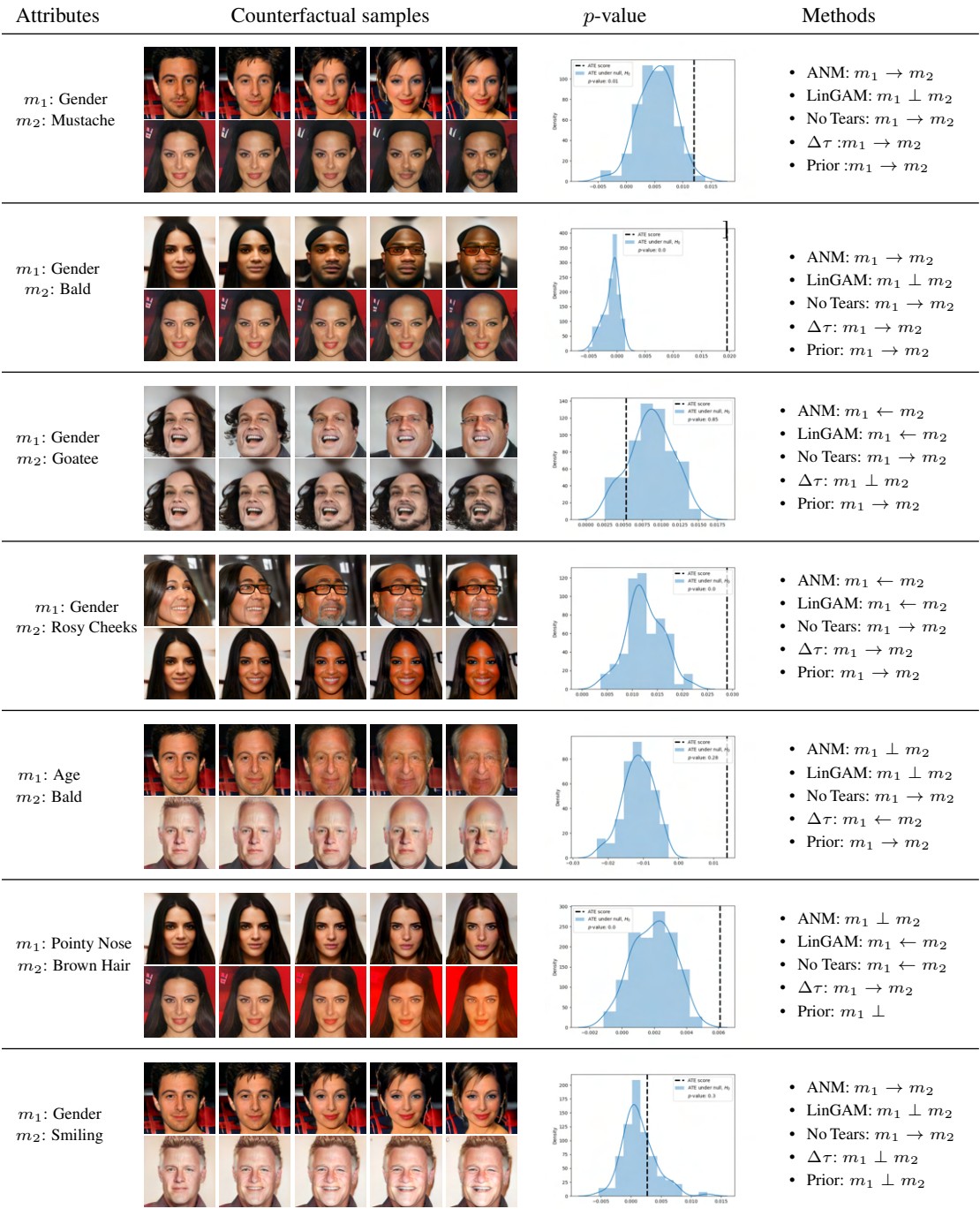

| Attributes | Counterfactual samples | $p$-value | Methods |
|---|---|---|---|
| $m_1$: Gender
$m_2$: Mustache | | | • ANM: $m_1 \rightarrow m_2$
• LinGAM: $m_1 \perp m_2$
• No Tears: $m_1 \rightarrow m_2$
• $\Delta\tau$: $m_1 \rightarrow m_2$
• Prior: $m_1 \rightarrow m_2$ |
| $m_1$: Gender
$m_2$: Bald | | | • ANM: $m_1 \rightarrow m_2$
• LinGAM: $m_1 \perp m_2$
• No Tears: $m_1 \rightarrow m_2$
• $\Delta\tau$: $m_1 \rightarrow m_2$
• Prior: $m_1 \rightarrow m_2$ |
| $m_1$: Gender
$m_2$: Goatee | | | • ANM: $m_1 \leftarrow m_2$
• LinGAM: $m_1 \leftarrow m_2$
• No Tears: $m_1 \rightarrow m_2$
• $\Delta\tau$: $m_1 \perp m_2$
• Prior: $m_1 \rightarrow m_2$ |
| $m_1$: Gender
$m_2$: Rosy Cheeks | | | • ANM: $m_1 \leftarrow m_2$
• LinGAM: $m_1 \leftarrow m_2$
• No Tears: $m_1 \rightarrow m_2$
• $\Delta\tau$: $m_1 \rightarrow m_2$
• Prior: $m_1 \rightarrow m_2$ |
| $m_1$: Age
$m_2$: Bald | | | • ANM: $m_1 \perp m_2$
• LinGAM: $m_1 \perp m_2$
• No Tears: $m_1 \rightarrow m_2$
• $\Delta\tau$: $m_1 \leftarrow m_2$
• Prior: $m_1 \rightarrow m_2$ |
| $m_1$: Pointy Nose
$m_2$: Brown Hair | | | • ANM: $m_1 \perp m_2$
• LinGAM: $m_1 \leftarrow m_2$
• No Tears: $m_1 \leftarrow m_2$
• $\Delta\tau$: $m_1 \rightarrow m_2$
• Prior: $m_1 \perp$ |
| $m_1$: Gender
$m_2$: Smiling | | | • ANM: $m_1 \rightarrow m_2$
• LinGAM: $m_1 \perp m_2$
• No Tears: $m_1 \rightarrow m_2$
• $\Delta\tau$: $m_1 \perp m_2$
• Prior: $m_1 \perp m_2$ |

et al. (2020). Here, the original MNIST digits are modified to respect a causal structure whereby the stroke thickness of the digits is a cause to the brightness (i.e. $T \rightarrow I$). Specifically, thicker digits are brighter while thinner digits are dimmer under the prescribed causal graph. We train a powerful normalizing flow in Mintnet (Song et al., 2019) and interrogate the direction of causality—if any— between thickness and intensity by first projecting all test set samples to their corresponding latent vectors. Fig 4 shows qualitative examples of counterfactuals as well a $\Delta\tau$ curve as a function of $\alpha$. Here the generated counterfactuals reduce the thickness while maintaining the average intensity. As observed, the model generates digits that contain holes but to compensate for the drop in intensity

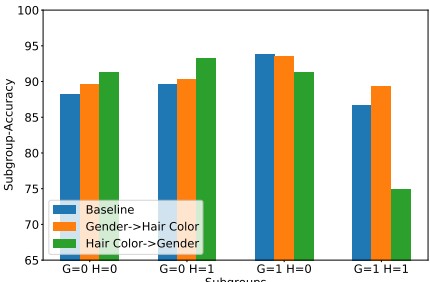

Figure 5: Subgroup classification accuracies for causal and anti-causal data augmentation and the baseline with no augmentation.

| Aug. Strategy | No Aug | $\mathbf{G} \rightarrow \mathbf{H}$ | $\mathbf{H} \rightarrow \mathbf{G}$ |
|---|---|---|---|
| Avg Group Train | 92.5 | 86.1 | 90.5 |
| Avg Group Val | 89.0 | 89.2 | 90.4 |
| Avg Group Test | 89.6 | **90.7** | 88.0 |
| Worst Group Train | 97.9 | 68.5 | 87.4 |
| Worst Group Val | 85.2 | 86.3 | 88.5 |
| Worst Group Test | 86.7 | **89.4** | 75.0 |

Table 2: Data Augmentation for Group Distribution-ally Robust Optimization on CelebA. In each DAG counterfactuals are generated by intervening on the causal parent while interpolating the effect attribute.

the model instead increases the intensity of the background pixels. Such generated samples are in line with expectations as we notice that the $\Delta\tau$ score indicates that the model has learned that thickness causes intensity and as such manipulating the causal variable propagates influence to the effect variable which manifests itself as an increase in background intensity.

**CelebaHQ**. We perform causal discovery over a pair of binary attributes in photo-realistic HD images as found in the CelebaHQ dataset (Karras et al., 2017). For our $G$ we employ a powerful generative model in StyleGAN2 (Karras et al., 2020) which is pretrained on the FlickrFacesHQ (Karras et al., 2019). To align the latent spaces between our desired causal discovery and FlickrFaces we finetune the pretrained model using $10\%$ of CelebaHQ. In table 1, we report the direction of causality found by each of the baselines and our approach using $\Delta\tau$. For each DAG considered we also generate counterfactual samples by manipulating one attribute and qualitatively observing downstream effects on the other. To diagnose independence between attributes, we also plot a histogram of $\Delta\tau$ scores under 100 random latent projections which allow us to reject the null hypothesis if the true $\Delta\tau$ (black line) is significant. In the absence of ground truth labels for causal associations, we posit a plausible causal relationship between each pair of attributes (e.g. Gender $\rightarrow$ Mustache) which we accept as a non-controversial biological prior. We observe that $\Delta\tau$ in all but one case—unlike each baseline —is consistently capable of diagnosing the causal relationship implicit to $G$. For example, for the causal relationship: Gender $\rightarrow$ Mustache, we observe counterfactual samples where gender (top row) is manipulated—i.e. from male to female–the mustache attribute also changes. In contrast, when the effect variable (bot row) is manipulated there is a reduced impact on the perceived gender. We also consider three causal discovery baselines: 1. LiNGAM (Shimizu et al., 2006) 2. Additive Noise Model (Hoyer et al., 2008), and 3. DAGs with No Tears (Zhang et al., 2018). For a fair comparison, and to be able to make causal statements about a generator as opposed to observed data, the causal discovery baselines operate on the class probabilities learned by the proxy classifiers described in §3.3 (see §B for further information on implementation details).

### 4.3 CAUSAL DATA AUGMENTATION

We now turn to a practical downstream application of generative modeling in the data augmentation of rare groups within a supervised learning task. Specifically, we consider the experimental setup in Sagawa et al. (2019) and perform group Distributionally Robust Optimization (DRO) over rare groups in CelebA. For a fair comparison with Sagawa et al. (2019) we choose to classify a person's hair color (**H**) which maybe is causally related with gender (**G**) (Hysi et al., 2018). In this case, the smallest group are blonde males that constitute only $\frac{1387}{162770} = 0.85\%$ of training examples. We use a finetuned StyleGAN2 to augment the total number of blonde males by a factor of 10 in two separate ways. First, we manipulate the hair color of all non-blonde males which we hypothesize corresponds to changing only the effect variable. The second approach seeks to generate additional samples of blond males by manipulating the gender of blond females in CelebA. For both approaches, we generate images of dimensions $1024 \times 1024$ which are then downsampled via bilinear interpolation to a resolution of $224 \times 224$ which is then consumed by a Resnet50 classifier.

In table 2, we report train, validation, and test accuracies on average across all subgroups and the worst-group (Blonde Males, i.e., **G**=1, **H**=1), while in figure 5 we report all subgroup accuracies for both the baseline approach and with data augmentation with $\alpha$ determined using the validation set. Additionally, we perform early stopping using the worst subgroup validation accuracy for all approaches. Fig 1, illustrates samples when gender for blonde females are manipulated to males (the

causal variable) versus manipulating the hair color of non-blonde males to blonde. Qualitatively, it is clear that if the end goal is to generate more blonde males the generative model must manipulate the effect variable. Quantitatively, we compute a bootstrapped FID score over $10$ trials for each setup's generated samples, using their respective against the true distribution of blonde males. We find that the FIDs for the counterfactuals in the inferred causal direction ($\mathbf{G} \to \mathbf{H}$) improve over the anticausal direction by $19.5\%$ revealing that using $G$'s learned causal relationship yields semantically better samples. Finally, we find that augmentation by manipulating the effect, hair color, in this case, leads to a $2.7\%$ gain in test accuracy in comparison to a $-1.6\%$ decrease when manipulating gender. Moreover, we observe the largest drop in test worst group accuracy when we generate synthetic samples without respecting the implicit causal associations in $G$, with classification accuracy dropping to $75\%$—serving as an important example for the need for a causal understanding of $G$.

## 5    RELATED WORK

CAGE is conceptually related to several methods, both within the generative modeling literature as well as the causal inference literature in general. With regards to the generative models, there is a rich line of work exploring the use of causal models to learn disentangled representations as advocated by Schölkopf et al. (2021) and Gresele et al. (2021). Both Sauer & Geiger (2021) and Besserve et al. (2018) leverage to notion of *independent mechanisms*, encoded as structural equation models, to obtain disentangled generative models which are subsequently employed to generate counterfactual samples. Pawlowski et al. (2020) propose a framework which incorporates deep probabilistic models as components within a structural equation model. In contrast with this work, CAGE instead focuses on quantifying when such causal biases have been internalized by a generator. In this sense, CAGE can be seen as solving the reverse problem to CausalGAN (Kocaoglu et al., 2017), which trains generative models such that they are consistent with a given causal graph.

Concurrently, there is a body of related work which leverages deep generative models to solve causal inference tasks. Louizos et al. (2017) and Yoon et al. (2018) employ generative models to estimate individual treatment effects by generating counterfactuals under alternative treatment regimes. Tran & Blei (2017) and Wang & Blei (2019) synthesize ideas from causality and generative modeling to address challenges associated with unobserved confounding. Finally, deep generative models have also been used for causal discovery tasks, examples include Kalainathan et al. (2018); Lopez-Paz & Oquab (2016) and Khemakhem et al. (2021), which is in sharp contrast with this work which focuses on uncovering causal structure within $G$, as opposed to the training data.

## 6    CONCLUSION

We proposed CAGE, a framework for inferring cause-effect relationships within the latent space of deep generative model via geometric manipulations. CAGE is well suited to a wide family of modern deep generative models trained on complex high-dimensional data and does not require any altering of the original training objective nor hard to obtain counterfactual data. Empirically, we find CAGE reliably extracts correct cause-effect relationships in controlled settings like MorphoMNIST. On high-resolution image data such as CelebAHQ, CAGE reveals—occasionally contrary to a biological prior—cause-effect relationships that are best supported by generated counterfactual samples. Finally, using causal insights supplied by CAGE we also perform guided data augmentations and power a supervised distributionally robust optimization task, obtaining a $2.7\%$ improvement over the state-of-the-art in rare subgroup classification.

**Limitations**. Despite CAGE's broad applicability to deep generative models, one important limitation is the requirement of a latent space. Fully observed generative models (e.g., autoregressive modes) do not have an explicit latent space and we will need to adapt their internal representations for probing via CAGE. In addition, the notion of GATE in CAGE, while being novel and tested rigorously empirically, does not entirely follow the same analysis as ATE. We implicitly borrow some key assumptions regarding unconfoundedness and introduce new ones such as linear separability in semantic attributes of the latent space in designing our estimator. These assumptions may not always hold in practice and future work can investigate both theoretically and empirically if and when mitigation strategies are needed. Finally, as we consider only binary attributes in this work, the extension of our framework to richer non-binary attributes and more than two variables (as in structural causal models) is an interesting direction for future work.

## ETHICS STATEMENT

As deep generative models grow in their practicality and deployment in high-stake applications, it is important to take a critical view of the inherent biases implicitly encoded in these models (Sheng et al., 2019; Choi et al., 2020; Bender et al., 2021). Our framework allows for a principled study of such biases through a causal analysis, potentially paving the way to mitigate the undesirable harms of deep generative models.

For example, our approach highlights important failings of the generative model itself—e.g. when cause-effect relationships are inconsistent with biological priors for human subjects or with physical relationships for scientific applications. In such cases, the use of these models in any high-stakes downstream application can be very costly without any intervention. Examples range from the current use of generative models for accelerating scientific discovery as well as societal applications in training fair classifiers. Ironically, in all such applications, the use of generative models can worsen the problem at hand by introducing new blindspots through the use of generated data that does not conform to the desired cause-effect relationship. Explicitly inferring these failings provides researchers and practitioners with an opportunity to develop algorithms and evaluation metrics that go beyond the current focus on visual fidelity or likelihoods as well as derive better data acquisition strategies for finetuning generative models.

On the other hand, we acknowledge that like many other efforts in generative modeling, our endeavors for better understanding these models can potentially be exploited by malicious actors for generating fake content, commonly known as Deepfakes (Korshunov & Marcel, 2018). Finally, causality has proven to be a highly powerful tool for formalizing important concerns of machine learning models with regards to fairness, robustness, and interpretability. However, as highlighted in our limitations, any causal framework including ours itself implicitly encodes some assumptions that may not hold in practice. Hence, researchers and practitioners should exercise due caution and appropriate judgment in interpreting the findings.

## REPRODUCIBILITY STATEMENT

Throughout the paper, we tried to provide as many details as possible in order for the results of the main paper to be reproducible. In particular, we provide implementation details in Appendix B for all the main models used. To ensure, the estimated $\Delta\tau$ values are significant we compute $p$-values under random projections in the latent space for CAGE. All the baseline approaches are additionally run over multiple trials. Furthermore, we provide additional generated samples in Appendix D to supplant all counterfactual samples provided in the main text. Finally, code to reproduce all experiments will be made public after the review process.

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

## A    EXTENDING AVERAGE TREATMENT EFFECTS TO GENERATIVE MODELS

For average treatment effects, we are interested in computing

$$\tau = \mathbb{E}_i \left[ Y_{m_c=1}(i) - Y_{m_c=0}(i) \right].$$

There are two important distinctions between the use of GATE in CAGE and a traditional ATE computation:

- First, as we are interrogating a generative model, we can observe *both* $Y_{m_c=1}(i)$ and $Y_{m_c=0}(i)$. This is a significant difference to standard ATE calculations, which can only observe one outcome.

- Second, we are interested in understanding implicit causal structure in the generator, $G$, the expectation over samples from the prior of $G$ as opposed to over samples from an observational dataset. This is an important distinction as it disambiguates the causal structure present in an observation dataset, which is the focus of traditional causal discovery methodology, with any causal structure implicit within the generator, which is our primary concern.

For unbiased estimation of ATE (a causal quantity) via statistical quantities, it is standard to assume unconfoundedness, positivity, and SUTVA. For the use of GATE in CAGE, we implicitly make further assumptions such as:

1. **Counterfactual parameterization**: The latent space of a generator, $G$, must be linearly separable with respect to the attributes of interest. This assumption is required as CAGE parameterizes the assignment of counterfactual treatments as linear manipulations over the latent space of a generator. We note that such an assumption has been shown to hold in practice for deep generative models (Denton et al., 2019; Ramaswamy et al., 2021).

2. **Complete support of the generator**: The generator, $G$, can generate high quality samples over the entire support of its latent space, $\mathcal{Z}$. This assumption is required to guarantee that linear manipulations over the latent space of a generator continue to produce high quality images, such that our resulting conclusions are not the result of e.g., image artefacts.

3. **Sufficiently expressive and accurate classifiers**: CAGE is premised on the use of probabilistic classifiers in order to quantify the presence/absence of an effect variable on counterfactual examples and thus compute treatment effects as specified in equations (5-6). We require accurate and calibrated classifiers (ideally, Bayes optimal) for computing GATE.

## B    MODEL DETAILS

For our experiments, whenever possible, we used the default settings found in the original papers of all chosen models. In particular, we used the default settings for both Mintnet (Song et al., 2019) and StyleGAN2 (Karras et al., 2020) which were pretrained on MNIST and FlickrFacesHQ respectively. For Mintnet we finetuned on a MorphoMnist dataset for 250 epochs using the Adam optimizer with default settings. Similarly, we also finetuned StyleGAN2 on CelebAHQ for 2000 iterations using 10% of CelebAHQ. Our synthetic experiments on the other required us to train a Masked AutoRegressive Flow (Papamakarios et al., 2017) that consisted of 10 layers with 4 blocks per layer. The Masked AutoRegressive Flow was trained for 5000 iterations using 5000 data samples.

For data-augmentation we train a ResNet50 classifier from scratch using weight decay of $1e-3$ as well as the default hyperparams for group DRO suggested in Sagawa et al. (2019). We also search over $\alpha$ in the range of $1-20$ while perorming early stopping using the worst subgroup validation accuracy. Note that the worst subgroup need not be blonde males throughout the entire optimization procedure.

### A NOTE ON BASELINE METHODS FOR CELEBHQ

We compared CAGE to well-established causal discovery algorithms when looking to infer structure over a deep generative model, $G$. As noted in §4.2, the causal discovery baselines considered operate over the class probabilities output by probabilistic classifiers trained as §3.3. We note that while it would be possible to learn causal structure using baseline methods, such as LiNGAM, over

the metadata (e.g., hair color and gender), this would not necesaly provide any insights into the causal structure implicit within the generator, $G$. For this reason, we instead focus on applying causal discovery methods over the output of proxy classifiers.

## C  ADDITIONAL BASELINE EXPERIMENTS

In this section we perform a series of ablation studies that stress test our framework CAGE under various experimental settings.

**Hard vs. Soft**. As we have access to ground truth labels it is tempting to consider whether using hard labels as opposed to the classifiers probability is better suited to computing our $\Delta\tau$ metric. In the table below we repeat our causal discovery experiment over CelebAHQ. As observed, all baselines provide unreliable estimates to the causal relationship between variables when compared to the main table which is computed using soft labels. Finally, we found it useful to assign soft labels provided by the classifiers used in the ATE computations for all augmented images during group DRO training.

| Method | ANM | DAGs with No Tears | LinGAM |
|---|---|---|---|
| Gender, Mustache | $\perp$ | $\longrightarrow$ | $\perp$ |
| Gender, Bald | $\perp$ | $\perp$ | $\perp$ |
| Gender, Goatee | $\perp$ | $\perp$ | $\perp$ |
| Gender, Rosy Cheeks | $\perp$ | $\longleftarrow$ | $\perp$ |
| Age, Bald | $\perp$ | $\longleftarrow$ | $\longrightarrow$ |
| Pointy Nose, Brown Hair | $\perp$ | $\longleftarrow$ | $\longrightarrow$ |
| Gender, Smiling | $\perp$ | $\perp$ | $\perp$ |

**Baselines without using CounterFactuals**. We now turn to the use of Counterfactuals when computing our baseline scores. Specifically, we attempt causal discovery purely using observation data and labels with both hard and soft labels, which is in contrast to the main paper which considered baselines which had access to classifier probabilities on counterfactual data. The table below shows this result of this ablation.

| Method | ANM | DAGs with No Tears | LinGAM |
|---|---|---|---|
| Gender, Mustache | $\longrightarrow$ | $\longrightarrow$ | $\perp$ |
| Gender, Bald | $\longrightarrow$ | $\longrightarrow$ | $\perp$ |
| Gender, Goatee | $\longleftarrow$ | $\longrightarrow$ | $\perp$ |
| Gender, Rosy Cheeks | $\longleftarrow$ | $\longrightarrow$ | $\perp$ |
| Age, Bald | $\perp$ | $\longrightarrow$ | $\perp$ |
| Pointy Nose, Brown Hair | $\perp$ | $\longleftarrow$ | $\perp$ |
| Gender, Smiling | $\longrightarrow$ | $\longrightarrow$ | $\longrightarrow$ |

## D  ADDITIONAL GENERATED SAMPLES

We now provide additional qualitiative examples of the generated counterfactuals on the same set of DAG's as considered in the main text. In the first row of all settings we generate counterfactuals by manipulating the first attribute variable (e.g. Gender) while the second row corresponds to counterfactuals where the second attribute is manipulated. The third row can either be the first or second attribute depending on context. We note that the generated samples may expose unintended biases that may be present in CelebAHQ and can be learned by the generator. While CAGE does not introduce any new sources of biases it should be noted that generating counterfactuals in this manner can further exacerbate these initially hidden biases. As a result, CAGE can serve as an important aid in empirically illuminating important ethical concerns that should be taken into account for any purposeful uses of generative models.

**Gender, Mustache**.

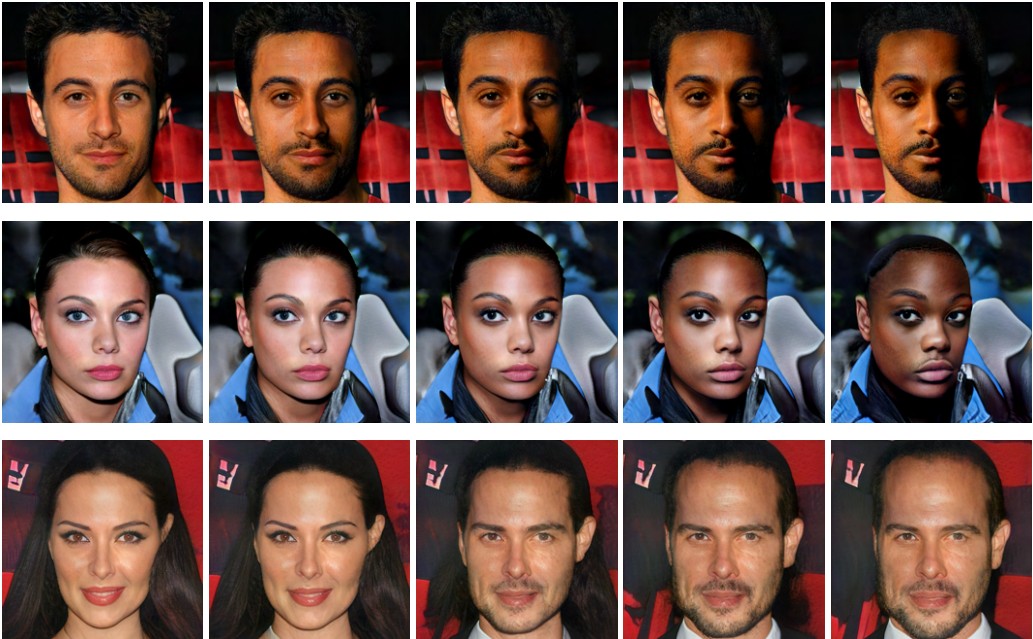

We consider generating samples by either manipulating the mustache attribute (top two rows) and then manipulating gender. As we can see, when the Gender is male we are able to semantically preserve the gender as we increase the presence of mustache (top row). Similarly, we find that when the gender is female we are still able to semantically preserve the (subjective) gender as we manipulate the mustache attribute (middle row). Finally, in the third row we change the gender of from female to male and notice that although the mustache attribute was not explicitly manipulated it appears as we manipulate gender more aggressively to male (bottom row). These results suggests that the generative models learned causal relationship is that *gender causes mustache*.

**Gender, Goatee**.

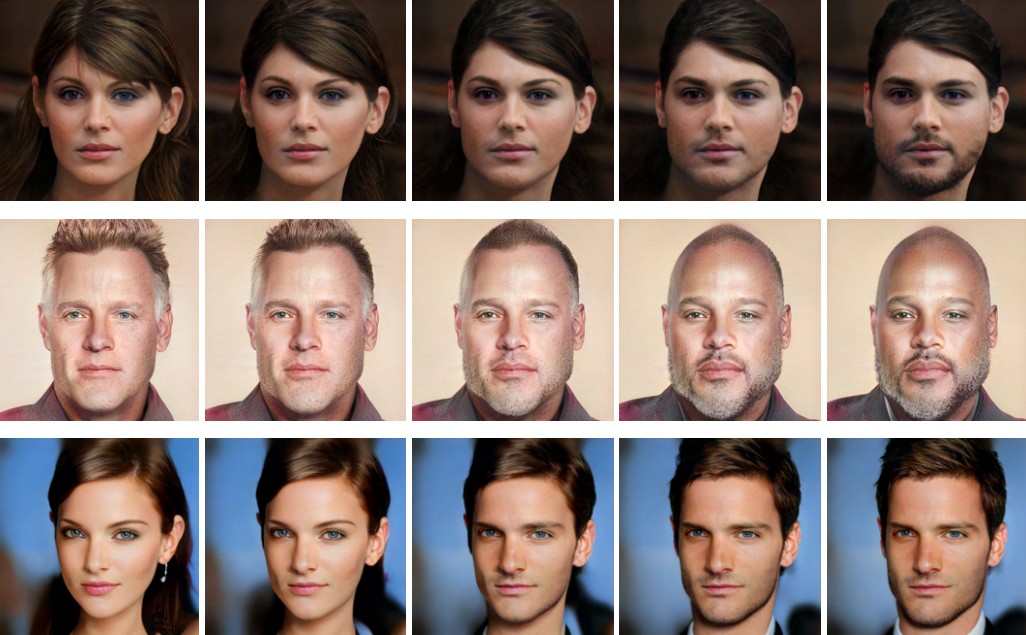

We consider generating samples by either manipulating the goatee attribute (top two rows) and manipulating gender. In the first row we see that manipulating the goatee attribute leads to a visual and subjective change the samples perceived gender (top row). Similarly, we find that when the gender is already male increasing the presence of a goatee does not reverse the gender to female as expected (middle row). Finally, in the third row we change the gender of from female to male and notice that there is no increase in goatee in the generated samples. These results suggests that the generative models learned causal relationship is that *gender is independent of goatee*.

**Gender, Smiling**.

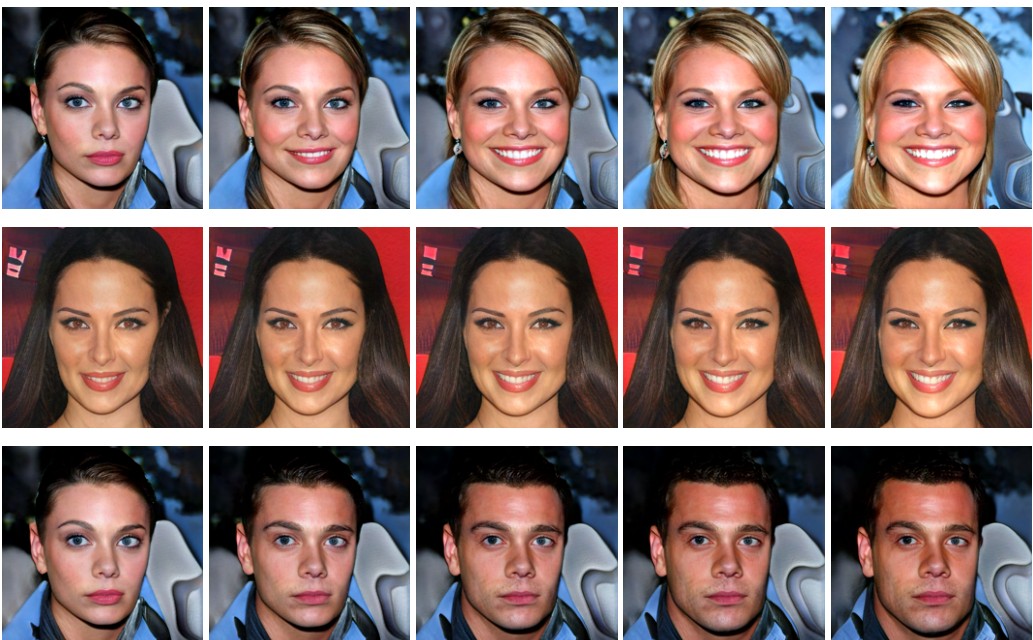

We consider generating samples by either manipulating the smiling attribute (top two rows) and manipulating gender. In the first two rows we see that increasing the smiling attribute does not cause a (subjective) change the samples perceived gender. Conversely, in the last row manipulating a samples gender from female to male preserves the smiling attribute (or lack thereof) in the generated samples. These results suggests that the generative models learned causal relationship is that *gender is independent of smiling*.

**Age, Bald**.

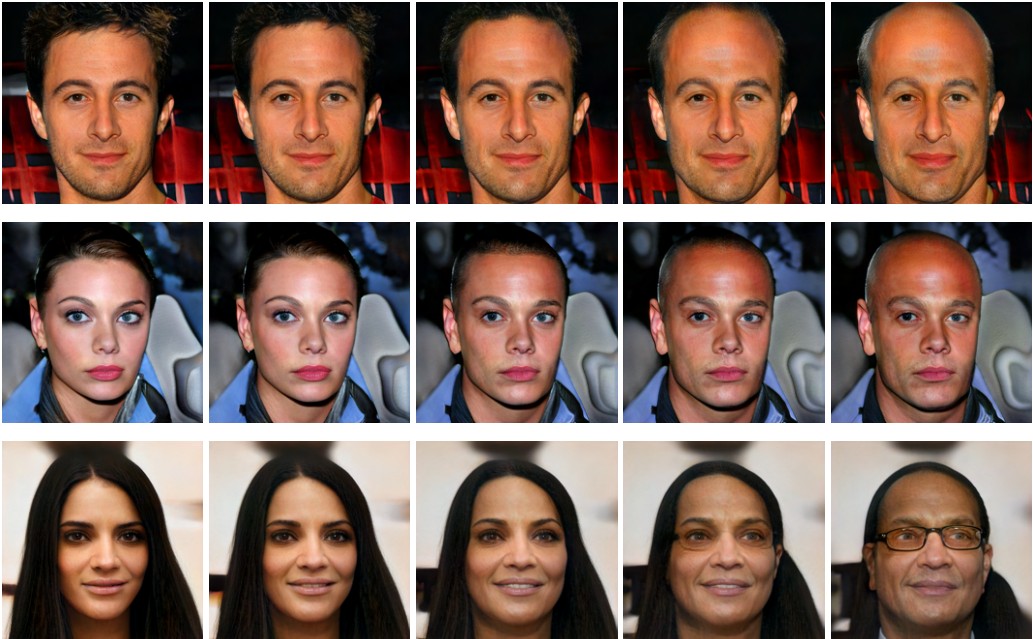

We consider generating samples by either manipulating the baldness attribute (top two rows) and then manipulating age (last row). In the first two row we see that increasing the baldness attribute does considerably change the age of the generated sample. On the other hand, in the last row we increase the age of the candidate and observe that while the baldness attribute increases it is not the extent as samples found in the first two rows. These results suggests that the generative models learned causal relationship, in contrast to our biological prior, is that *baldness is the cause of age*.

**Pointy Nose, Brown Hair**.

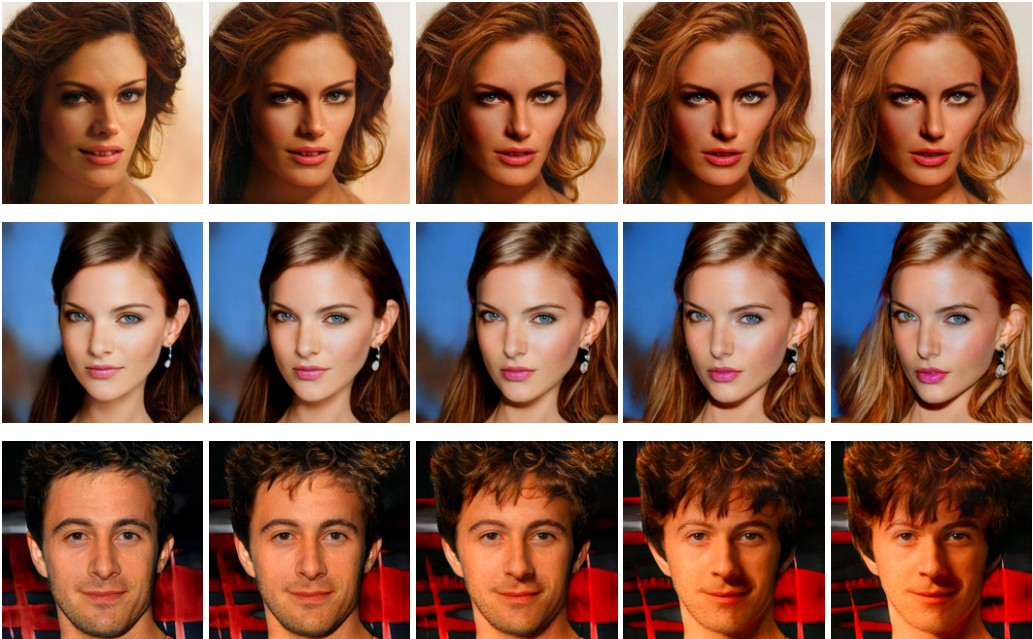

We consider generating samples by either manipulating the pointy nose attribute (top two rows) and manipulating brown hair. In the first two rows we see that increasing the pointy nose attribute does cause a change in the samples hair color from dark brown to more blonde. Conversely, in the last row manipulating a samples hair color to be more brown does not appear to significantly change the pointy nose attribute in the generated samples. These results suggests that the generative models learned causal relationship is that *pointy nose causes brown hair*.

**Gender, Rosy Cheeks**.

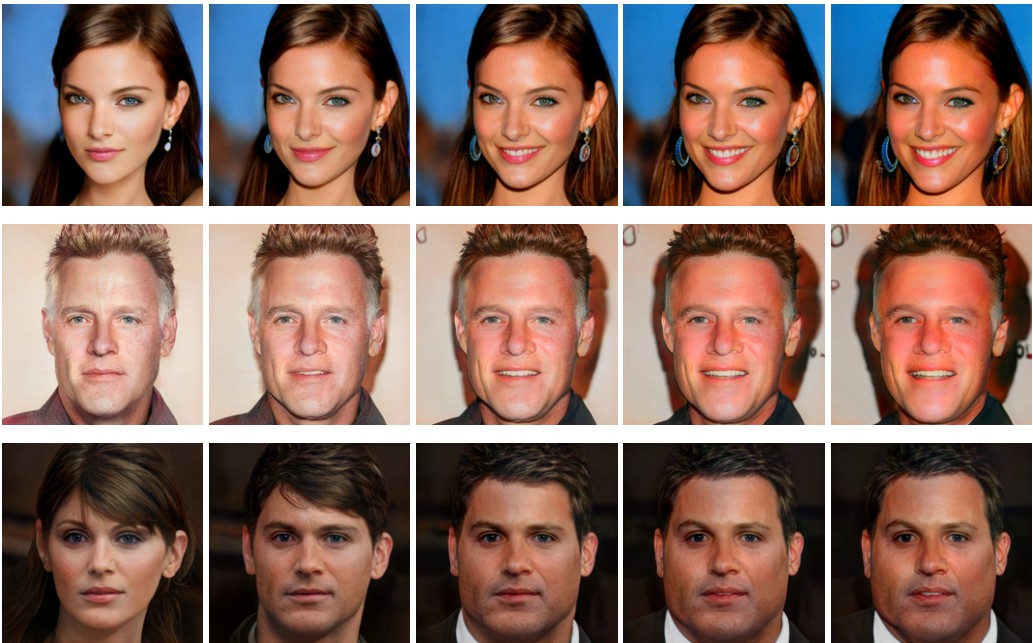

We consider generating samples by either manipulating the rosy cheek attribute (top two rows) and manipulating gender. In the first two rows we see that increasing the rosy cheek attribute does not cause a change in the samples (subjective) gender—i.e. both females and males retain their gender as we increase their rosy cheeks. On the other hand, in the last row manipulating a samples gender from female to male appears to slighltly decrease the rosy cheek attribute in the generated samples. These results suggests that the generative models learned causal relationship is that *gender causes rosy cheeks*.

**Gender, Blonde Hair**.

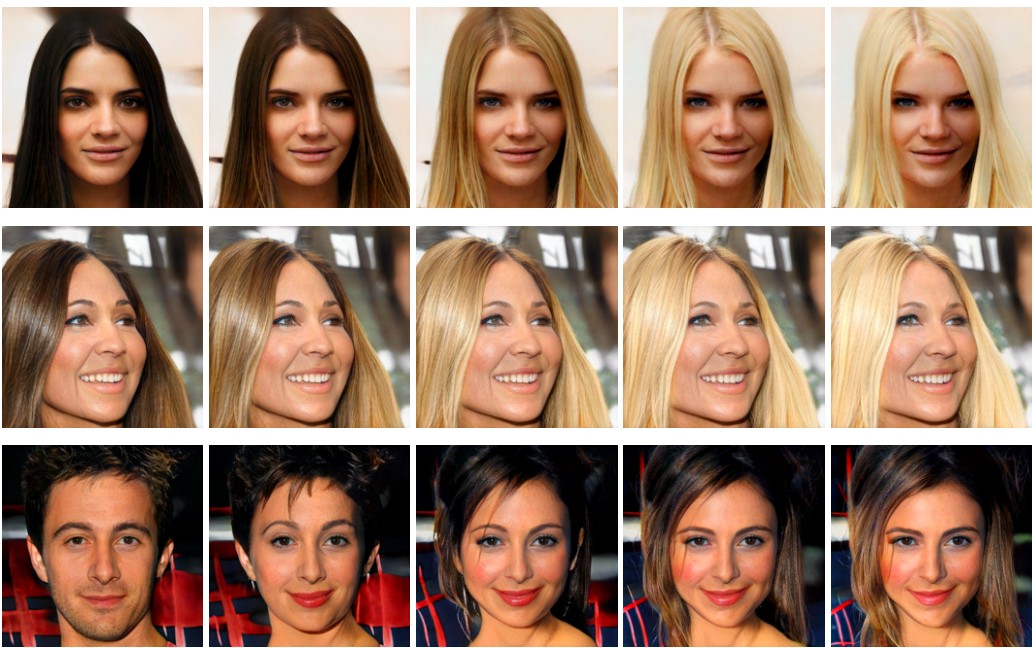

We consider generating samples by either manipulating the blonde hair attribute (top two rows) and manipulating gender. As observed in the first two rows increasing the blondeness of a samples hair does not change the perceived gender from female to male. In sharp contrast, the last row we see that manipulating the gender from male to female leads to the samples hair color changing towards being more blonde. These results suggests that the generative models learned causal relationship is that *gender causes blonde hair*.

# E MMD DISTANCE BETWEEN CONDITIONAL AND COUNTERFACTUAL DISTRIBUTIONS

In this appendix we plot the Maximum Mean Discrepancy distance between the conditional distribution on the latent space of an attribute and the counterfactual distribution realized through our CAGE framework. Specifically, we plot the MMD distance as a function of the interpolant strength $\alpha$ and we observe that the two distributions grow further apart the further we interpolate past the linear classifiers hyperplane as expected.

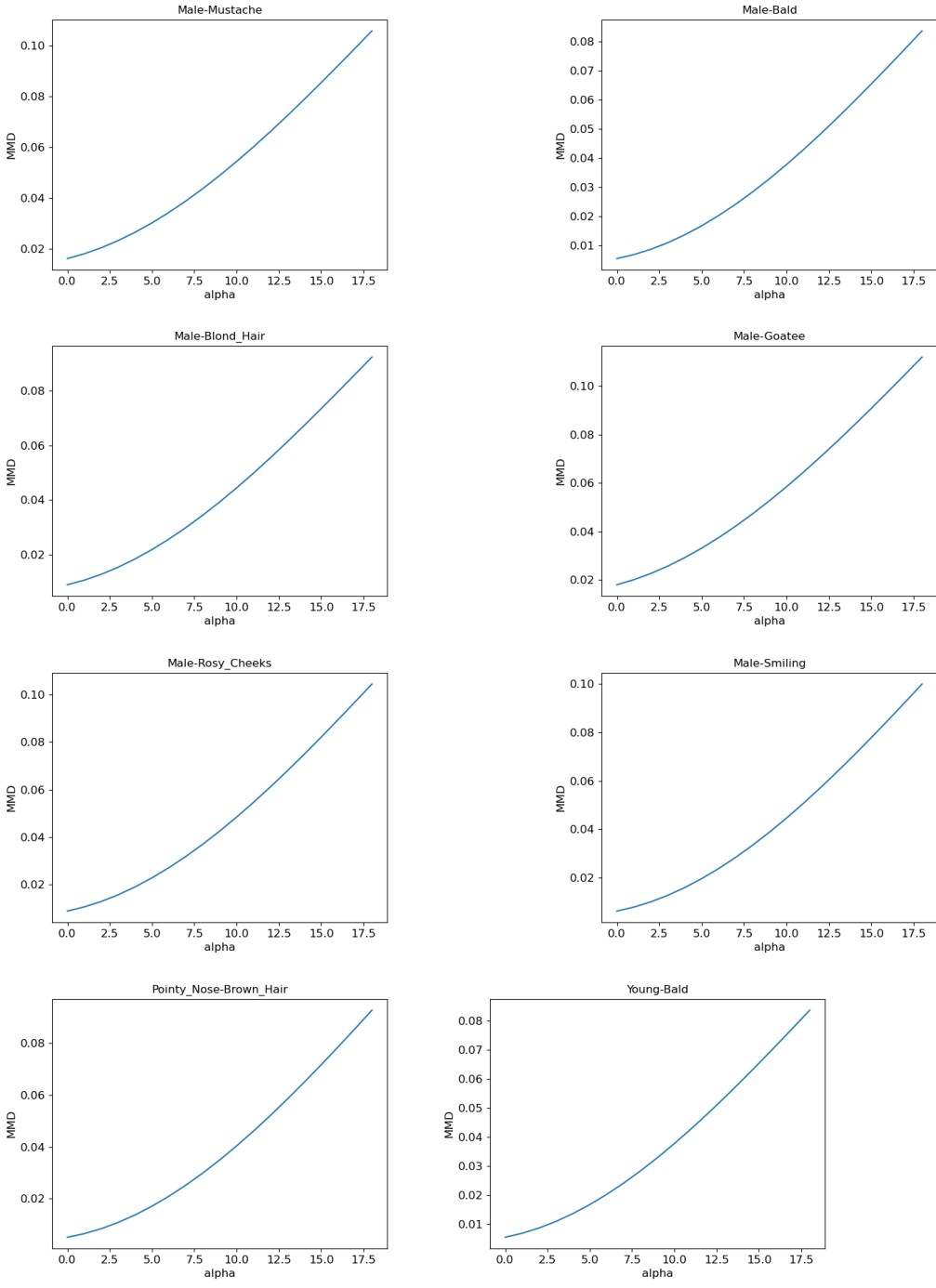

