# OpenReview forum: "CAGE: Probing Causal Relationships in Deep Generative Models"
_ICLR.cc/2022/Conference — ICLR 2022 Submitted_

### Official Review · Reviewer_xP5d · 2021-10-26

**Correctness:** 2
**Technical Novelty And Significance:** 3
**Empirical Novelty And Significance:** 3
**Recommendation:** 3
**Confidence:** 4

**Main Review:**

PROS:

The goal of the paper is fresh, I am not aware of other works trying infer causalities from the latent space in a similar way as done here.

The use of the method for data augmentation seems promising.

The presentation is clear and technically correct, although some more theoretical justification for calling the relationships causal, and not just statistical, would be beneficial.

CONS/QUESTIONS:

My main concern is that this paper appears to be more about the statistical dependences of two variables in the latent space - and while some of the results seem promising - it is not clear to what extent these findings have to do with causality, and to what extent they are just conditional probabilities. It is quite possible that causality leaves some trace that can be detected in the latent space, but it is not clear what that trace could be.

I would suggest the following, at least, to improve this:
1) Visualizations of the latent space and the decision boundaries to provide intuition about what's happening in the latent space, and where does the causal signal come from.
2) Many more experiments with pairs of variables, including (but not limited to) such variables that are correlated but not causally related. This would allow the reader to see whether the method can truly distinguish causality from association.

Permutation removes not only causality, but any association altogether. Therefore, it can not be used to test the statistical significance of "causality".

The results regarding pairs of variables are too limited to convince a reader about the ability of the method to infer causal relationships in general (and not just statistical). Many of the pairs appear to be of the kind where the putative effect is present in a subset of data items where the putative cause is present (e.g. gender->mustache, those who have mustache are a subset of male) - I'm wondering what role this asymmetry plays in the results? With the current results, the reader is left to wonder if there are other similar asymmetries that could lead to a (potentially non-causal) signal that can be picked by the method?


**Summary Of The Paper:**

The goal of this paper is to inspect causal relationships learned by a generative model. In particular, the paper studies how moving in the latent space with respect to a putative cause variable (e.g. age) affects the probability of the putative effect variable (e.g. baldness). This is done by training two linear classifiers in the latent space, one for each variable. The counterfactual for a given cause variable and datapoint is defined by changing the location of the data point in the latent space in the direction perpendicular to the linear classification boundary for the cause variable. Then, the paper inspects what happens to the putative effect variable, using its respective decision boundary. A null distribution is defined using permutation. The usefulness of techniques is demonstrated for inferring the causal relationships (including direction) for pairs of variables and for generating data augmentations for robust classification.


**Summary Of The Review:**

The paper presents interesting ideas to investigate the relationships between variables learned by a deep generative model, but to make the claim that these relationships are causal, more theoretical justification, intuition, and experiments would be needed.

POST-REBUTTAL UPDATE:

I thank the authors for their attempt to improve the paper. Nonetheless, I don't find my original concerns resolved, nor answers to my suggestions. Especially the framing of the approach using causal concepts remains obscure and unconvincing.

As a reply to the authors' specific question on permutation: if two variables are strongly associated but not causal, and the method incorrectly detects a causal signal between the variables (based on the presented experiments it is not inconceivable that this may happen in some cases), then the permutation test will highlight this as highly significant (because the association is removed in the permuted dataset).

---

> ### Author Response · Authors · 2021-11-17
> **Response to Reviewer xP5d Part 1/2**
>
> We thank the reviewer for their helpful comments and feedback on our work. In particular, we appreciate that the reviewer felt that the goal of the paper was “fresh”,our proposed methodology seems promising, and that the presentation was “clear” and “technically correct”. We now address the key questions raised by the reviewer below.
>
>
> **Q. Causal signal and traces in latent variable models beyond statistical correlations.**
>
> Great question! The key element for deriving a causal signal in CAGE is our counterfactual generation strategy (step 3, page 4). This allows us to explicitly simulate a counterfactual via a geometric operation on the treatment variable --- there isn’t any statistical inference query (eg, conditional probabilities) that can achieve the same effect. For defining our geometric operation in latent space, we lean on numerous empirical studies in generative models. It is also important to note that while we do parameterize counterfactuals in the latent space of a deep generative model, ultimately we are interested in understanding implicit causal structure within the generative model itself. That is, a key component of our proposed methodology is to understand the effects of our interventions in the latent space on the corresponding samples produced by the generative model.
>
> As qualitative evidence, notice that many of the counterfactuals samples in the paper possess attributes that do not exist in the training dataset at all, e.g., females that have moustaches or are bald. Similarly, we do not observe such individuals when generated from the prior *without* any counterfactual manipulation. However, the fact that our counterfactual manipulation strategy can reliably generate such individuals is strong evidence that our counterfactual distribution is different from the statistical distribution of the model.
>
> Quantitatively, we can measure the Maximum Mean Discrepancy (MMD) distance between samples from the counterfactual distribution and the conditional distribution (i.e., obtained via restricting the samples to a particular subgroup) in Appendix E. The significant difference in the values between the counterfactual and conditional distribution suggests that our method makes inferences that go beyond statistical correlations.
>
>
> **Q. Experiments with pairs of variables, including (but not limited to) such variables that are correlated but not causally related.**
>
> We included the Gender and Smiling pair in our original experiments. Here, our biological prior suggests no causation but there are correlations in both the training dataset (14.2% males vs. 32.8% females are smiling) and in the generated examples (10.0% males vs. 36.0% females are smiling, with labels determined by a pretrained classifier). CAGE infers the two attributes to be independent, consistent with our biological prior. Also, as we note in the paper as well, our goal is to evaluate the causal relationships within a deep generative model. These relationships may or may not be reflected in the training data or our existing priors -- in fact, any evidence of inconsistency lends a non-trivial understanding of the generative model and control its use for downstream applications.  If the reviewer has specific pairs of attributes in mind for including, please let us know and we would be happy to include results in the remaining time!
>
>
> **Q. Permutation removes not only causality, but any association altogether. Therefore, it can not be used to test the statistical significance of "causality".**
>
>
> We would be grateful for a clarification on the basis for the above statement. Permutation-based tests of significance have been used for a long time in the causality literature, even dating as far as to the work Fisher (see e.g., [1, 2, 3]). The intuition in our case is to assert whether the estimated treatment effect is statistically significant. To determine whether this is the case, we first posit a distribution for treatment effects in the absence of causal associations (as the reviewer mentions, the use of permutations does indeed remove the causal link). Then, given an empirical approximation to the null distribution, we are able to determine whether an estimated generative treatment effect estimate is significant.
>
>
> Response Part 1/2

---

> > ### Author Response · Authors · 2021-11-17
> > **Response to Reviewer xP5d Part 2/2**
> >
> > **Q. Many of the pairs appear to be of the kind where the putative effect is present in a subset of data items where the putative cause is present.**
> >
> > We included pairs that do not have a strict subset relationship, e.g., age and bald -- both young and old people can be bald, gender and hair color -- both genders can have blonde hair.  For the pairs which have subset relationships, it is purely coincidental and we picked those pairs only because we have good biological priors on the direction of causation in such cases as a reference. In fact, we believe the very fact that our method can simulate counterfactuals even when they are not represented in the dataset is strong evidence of the success of our treatment effect estimation (and cannot simply be obtained via a conditioning operation on the statistical joint distribution). If the reviewer has specific pairs of attributes in mind for including, please let us know and we would be happy to include results in the remaining time!
> >
> > References
> > [1] Fisher, R. A. (1935). The Design of Experiments. Edinburgh: Oliver & Boyd.
> >
> > [2] Basse, Feller & Toulis, “Randomization tests of causal effects under interference", Biometrika (2019)
> >
> > [3] Aronow, Peter M., and Cyrus Samii. "Estimating average causal effects under general interference, with application to a social network experiment." The Annals of Applied Statistics 11.4 (2017): 1912-1947.
> >
> > Response Part 2/2

---

> ### Author Response · Authors · 2021-11-23
> **Kind Reminder to Respond to our Rebuttal**
>
> Dear Reviewer,
>
> We are grateful for our feedback and comments that allowed us to strengthen our paper. We would like to highlight that we have updated our manuscript and responded to all of your key comments below in our rebuttal. We hope our response was satisfactory and aids you in potentially reevaluating our paper in a favorable light. We would greatly appreciate a response from you and we are also happy to answer any remaining questions that you might have.

---

> ### Author Response · Authors · 2021-11-29
> **Re:Post-Rebuttal Update**
>
> We thank the reviewer for acknowledging our updated rebuttal today. The reviewer's main remaining concern appears to relate to distinguishing causality from correlation. This is indeed a difficult challenge, and one that all causal discovery algorithms must contend with.
>
> We note that the scenario the reviewer describes, where two variables are strongly associated but not causal, is one that falls under the framework of an unobserved confounder [1]. In Appendix A of our submission we outline the assumptions required for the ATE, a causal quantity, to be estimated via statistical quantities. Amongst the assumptions made is unconfoundedness, which is a standard assumption within the causal discovery literature (for example, here is a non comprehensive list of related works making the same assumption: [2, 3, 4, 5, 6, 7]).
>
> Furthermore, we note that our measure of causal direction is in fact a **difference in estimated ATEs** (treating each variable as the candidate causal variable). As such, in the scenario described by the reviewer, we would hope our measure of causal direction as defined in equation (7) would indeed be small relative to the variance of null distribution (obtained under permutation). In fact, as we mentioned in our rebuttal, we provide the example of Gender and Smiling where this is indeed the case: there is a correlation present in the dataset but CAGE does not infer any causal association. This can be seen visually in the final row of Table 1 and in the additional generated samples found in the appendix.
>
> Finally, we note that whilst the assumption of unconfoundedness is a widely employed theoretical assumption, in practice it can often be difficult (if not impossible) to determine if an unobserved confounder is present. For this reason we find the example provided with Gender & Smiling to be particularly compelling.  Even for the other attribute pairs, we followed up on the reviewer’s suggestion in comparing the conditional and counterfactual distributions. We added a new appendix E containing the MMD between these distributions. The plots are available at the links below for convenience. These results demonstrate that the conditional distribution is different in MMD distance with respect to the counterfactual distribution
>
> Links to MMD:
> https://ibb.co/Yhvt3p8
>
> https://ibb.co/V3Zg4Xf
>
> https://ibb.co/B4XFTKm
>
> https://ibb.co/sWd1VG9
>
> https://ibb.co/jG2Hk03
>
> https://ibb.co/kgyYnJG
>
> https://ibb.co/khZXCWv
>
> https://ibb.co/LzS6RQJ
>
> Please let us know if you have any further questions. While today is the last day for author responses, we’ll be happy to address them as soon as possible.
>
> References:
> 1. Pearl, Judea. Causality. Cambridge university press, 2009. Chapter 6.
> 2. Mooij, Joris M., et al. "Distinguishing cause from effect using observational data: methods and benchmarks." The Journal of Machine Learning Research 17.1 (2016): 1103-1204.
> 3. Pawlowski, Nick, Daniel C. Castro, and Ben Glocker. "Deep structural causal models for tractable counterfactual inference." Proceedings of 34th Int. Conf. Neural Information Processing Systems (2020)
> 4. Yoon, Jinsung, James Jordon, and Mihaela Van Der Schaar. "GANITE: Estimation of individualized treatment effects using generative adversarial nets." International Conference on Learning Representations. 2018.
> 5. Alaa, & van der Schaar. "Bayesian inference of individualized treatment effects using multi-task gaussian processes." NeurIPS (2017).
> 6. Athey, Susan, and Guido Imbens. "Recursive partitioning for heterogeneous causal effects." Proceedings of the National Academy of Sciences 113.27 (2016): 7353-7360.
> 7. Shalit, Uri, Fredrik D. Johansson, and David Sontag. "Estimating individual treatment effect: generalization bounds and algorithms." International Conference on Machine Learning. PMLR, 2017.

---

### Official Review · Reviewer_TgM5 · 2021-11-01

**Correctness:** 3
**Technical Novelty And Significance:** 1
**Empirical Novelty And Significance:** Not applicable
**Recommendation:** 3
**Confidence:** 4

**Main Review:**

I would like to thank the authors for the interesting work they proposed, and tried to explain my concerns below.

1. My first concern is that the authors try to probe causality in a trained model, instead from the data. The trained model may not represent the data well nor capture the causality in the true underlying process that generates the data. In fact, the styleGAN model used is not even designed to capture causality. It is unclear why probing causality in such a trained model is useful at all.

2. Even for a causal discovery model that aims to estimate causality for latent variables from the data, they often reply on heavy assumptions. Whether recovering the true parameter or the causal structure is possible is known as identifiability problem. I suggest that the authors to check some particular identifiable generative models, e.g., CausalGAN (Kocaoglu, Murat, et al. "Causalgan: Learning causal implicit generative models with adversarial training." arXiv preprint arXiv:1709.02023 (2017).), not “any existing pre- trained latent variable model” shown in the abstract. In addition, I recommend the authors read some papers relevant to identifiability in latent space, e.g., Khemakhem, Ilyes, et al. "Variational autoencoders and nonlinear ICA: A unifying framework." International Conference on Artificial Intelligence and Statistics. PMLR, 2020.

3. The term 'latent' variables used in this manuscript is misleading. My understanding is that the authors used StyleGAN which produces style vectors. Once produced, these vectors are observed, thus are not really latent. Discovery causal relationship among observed variables have been well studied in the existing literature.

4. I am curious about the definition on the equation in ‘step 3, page4’. In causality, do’ operator has strict mathematical definition, it is unclear whether ‘do’ operator can be replaced by the RHS in the equation in theory. Or what the different between the RHS and the true ‘do’ operator. What is more, by using this definition, is the true potential outcomes equal to the ‘potential outcomes’ obtained by the equation? This is a very strange equation, it would be better if the authors can show that the potential outcomes obtained by the equation is consistent with the true outcomes in theory.

5. Counterfactuals require to recover the exogenous noise variables first, which is different from intervention. I don't see the noise variables were recovered in the manuscript.

---Post-rebuttal review----
The counterfactual is different from intervention. What the authors did in the paper is intervention, but claimed as counterfactuals. The authors seem to point out that they followed "section 4.1 of [9]", where I didn't find the support. Counterfactual reasoning requires recovering the exogenous variables (a.k.a. disturbance/noise variables) first, and then performs intervention.

It remains unclear what is the benefit of probing causality in a pre-trained model which is not designed to capture causality.

I am happy with the rest of the responses, thus have lifted up the score from strong rejection to rejection.

**Summary Of The Paper:**

The authors propose to discover causality for a trained man-made model. This is very unusual, as normally we would be interested in discovering causality the true underlying process that generates the true data.

More specifically, they propose a framework called CAGE to estimate generative average treatment effects (GATEs) for probing cause-effect relationships in the so called latent space of deep generative models. They use the average treatment effects (ATEs) to infer causal directions for deep generative models' attributes and use CAGE to generate counterfactual data. They evaluate the ability of CAGE both synthetic (a variant of MNIST called MorphoMNIST) and high-resolution face dataset (CelebaHQ).


**Summary Of The Review:**

The task is not well motivated. There are several technical issues mentioned above. I think this paper is not ready to publish.

---

> ### Author Response · Authors · 2021-11-17
> **Response to Reviewer TgM5 Part 1/2**
>
> We thank the reviewer for the comments and feedback. Unfortunately, we believe that the reviewer might have misunderstood some fundamental aspects of our work, which we highlight below.
>
> **Motivation and setup**
> - The reviewer suggests that understanding causality over a synthetic model is “unusual” and suggests causal structure can only be understood in the context of the underlying data generating mechanism.
>
> - We agree with the reviewer that understanding causal structure over deep generative models is a novel avenue of research (indeed other reviewers have also highlighted this and we view this as an important contribution of our work). We also agree that typically applications of causal inference do relate to observational data. However, as we note in our manuscript (and also as highlighted by reviewer Ak2p), the increasing prevalence of deep generative models within machine learning applications (e.g., scientific discovery [1] and decision making [2]) mean it is imperative to develop frameworks through which to better understand such models. This is the overall motivation for our work.
>
> - The reviewer highlights that such a causal model may “not represent the data well nor capture the causality in the true underlying process that generates the data”. The issue of poor model fit is orthogonal to our work, and indeed there are methods available to test model fit (e.g., [3]). The second issue, relating to whether a deep generative model has captured the “true underlying causal process” is indeed one of the primary motivations of our work. However, we do not aim to make statements over the true data generative mechanism but rather over implicit causal structures instilled in a generative model during training. Our work is the first to propose a mechanism through which to understand such implicit causal structure present in deep generative models, and thus determine if and when they are in agreement or contradiction with what we may expect (e.g., due to a biological prior belief). Given our limited understanding of the theoretical properties of deep models, and their propensity to learn shortcuts [4], it is by no means guaranteed that deep generative models will learn the true causal structure present in data. As such, we feel the motivation for our work is valid.
>
> **Identifiability and relation to prior work**
> - The reviewer notes that causal discovery methods often rely on heavy assumptions and references causalGAN [5] and iVAE [6]. Unfortunately neither of these references look to address the same issues as we do in this work.
>
> - First, CausalGAN does not make any claims regarding identifiability or learning of the causal graph, and rather assumes causal structure over attributes is provided apriori. In Section 3.1 they write “This paper does not address the problem of learning the causal graph: We assume the causal graph is given to us”.
>
> - Second, iVAE does indeed propose an identifiable deep generative model (to achieve this the authors introduce several assumptions, such as assumption of conditionally factorial priors). However, such a model is only able to learn causal relationships over *observed* variables (see also [7]). Whereas the setup described in our work is fundamentally different, as we look to infer causal structure over attributes, $m_i$ (e.g., gender, smiling, etc) rather than over the observed variables (i.e., images). This makes our approach fundamentally different, as neither [6] nor [7] consider causal structure over attributes and instead focus exclusively over observations $X$.
>
> - Finally, we note that we do *not* make statements about the identifiability of latent variables in generative models (we agree with the reviewer that such a claim is impossible without introducing additional constraints). Instead, Section 3 outlines an extension of the Neyman-Rubin potential outcomes framework to generative models. Such a framework is focused on unbiased estimation of an average treatment effect, as opposed to identifiability of latent variables. We outline the assumptions required to achieve this in Appendix A.
>
> **Use of term ‘latent’**
> Throughout the manuscript we use the term latent variables to refer to variables which are not directly observed but rather inferred. It is important to note that inferred latent variables are contingent on both observed data as well as a corresponding inference model, and thus are fundamentally different from observational data.  We believe our use of the term latent is in line with the literature (see for example [8, Chapter 12 “Continuous Latent Variables”]).
>
> Response Part 1/2

---

> > ### Author Response · Authors · 2021-11-17
> > **Response to Reviewer TgM5 Part 2/2**
> >
> > **Definition of do operator and computation of counterfactuals**
> > The reviewer notes that the do operator has a precise definition within causal inference and its connection to step 3 of the manuscript is unclear. Part of the reason for this is that our work is rooted within the framework of Potential Outcomes and Rubin Causal Model. Within this framework, the do-operator is implicitly defined within the treatment assignment mechanism (see section 4.1 of [9]). One important benefit of our work is that, since we study generative models, we are able to obtain samples under both treatments, thereby alleviating the “missing data” problem typically associated with the potential outcomes framework. We note that this approach and language is also exploited in other publications (e.g., [10]).
> >
> > We thank the reviewer for their time and effort in reviewing our work and we hope the reviewer would kindly consider a fresh evaluation of our work given the main clarifying points outlined above.
> >
> >
> > References:
> > [1] Benjamin Sanchez-Lengeling and Alan Aspuru-Guzik. Inverse molecular design using machine learning: Generative models for matter engineering. Science, 361(6400):360–365, 2018.
> >
> > [2] Lili Chen, Kevin Lu, Aravind Rajeswaran, Kimin Lee, Aditya Grover, Michael Laskin, Pieter Abbeel, Aravind Srinivas, and Igor Mordatch. Decision transformer: Reinforcement learning via sequence modeling. arXiv preprint arXiv:2106.01345, 2021
> >
> > [3] Salimans, T., Goodfellow, I., Zaremba, W., Cheung, V., Radford, A., & Chen, X. (2016). Improved techniques for training gans. In Advances in Neural Information Processing Systems (pp. 2234–2242).
> >
> > [4] Geirhos, Robert, et al. "Shortcut learning in deep neural networks." Nature Machine Intelligence 2.11 (2020): 665-673.
> >
> > [5] Kocaoglu, Murat, et al. "Causalgan: Learning causal implicit generative models with adversarial training." arXiv preprint arXiv:1709.02023 (2017).
> >
> > [6] Khemakhem, Ilyes, et al. "Variational autoencoders and nonlinear ica: A unifying framework." International Conference on Artificial Intelligence and Statistics. PMLR, 2020.
> >
> > [7] Monti, Ricardo Pio, Kun Zhang, and Aapo Hyvärinen. "Causal discovery with general non-linear relationships using non-linear ica." Uncertainty in Artificial Intelligence. PMLR, 2020.
> >
> > [8] Bishop, C. M. Pattern recognition and machine learning. (2007).
> >
> > [9] Pearl, Judea. "Causal inference in statistics: An overview." Statistics surveys 3 (2009): 96-146.
> >
> > [10] Yoon, Jinsung, James Jordon, and Mihaela Van Der Schaar. "GANITE: Estimation of individualized treatment effects using generative adversarial nets." International Conference on Learning Representations. 2018.
> >
> > Response Part 2/2

---

> ### Author Response · Authors · 2021-11-23
> **Kind Reminder to Respond to our Rebuttal**
>
> Dear Reviewer,
>
> We appreciate all of your comments and feedback provided in your review of our work. We wish to highlight that we have updated our manuscript and responded in detail to each point in our rebuttal below. We believe we have successfully addressed the chief points of contention and we ask you for a renewed evaluation of our paper with the rebuttal and updated manuscript as context. We are also more than happy to answer any remaining questions you might have that could alleviate this process further.

---

> ### Author Response · Authors · 2021-11-30
> **Re: Post-rebuttal review part 1/2**
>
> We thank the reviewer for acknowledging our revision & rebuttal as well as for raising their score. We hope that by clarifying the few remaining issues we can encourage the reviewer to raise their score again.
>
> It appears that the remaining points of contention center around semantics (counterfactuals versus interventions) and motivations for our work. We respond to each of the concerns below.
>
> We agree with the reviewer that the counterfactual is indeed different from an intervention and does require first recovering the latent exogenous variables. Thereafter, both counterfactuals and interventions proceed in the same manner as noted by the reviewer. In Step 1 of our proposed framework, we do indeed infer the associated latent representation, z(i), associated with an image, x(i), by inverting the deep generative model in question. This latent representation, z(i), is indeed the exogenous variable the reviewer is referring to as **it is conditioned on the image x(i)**. As an intuitive example, our framework looks to take an image (e.g., a blond woman) and posits the question: does changing the gender of the image change the hair color? This is clearly a counterfactual query and indeed the images we generate using our framework are counterfactuals because they continue to resemble the original image whilst changing the target attribute (in this case gender). If we were simply performing interventions (and not counterfactuals) we would just take random samples over the latent space (i.e., sample from the support of Z) and perform Step 3 and as a result we would see any resemblance/similarity in generated samples.
>
> Moreover, we note in our manuscript that for invertible models, such as flows, inference of z(i) can be achieved by inverting the flow models (e.g., as described in [1,2]). In the case of GANs, we can employ a separate encoder for this purpose - we are also not the first to propose this for counterfactual inference, see for example [3] who employ GANs to generate counterfactuals as well as [4], whose framework and setup closely matches what we describe in steps 1-3. We would be happy to update the description of Step 1 of our proposed framework to highlight that it is indeed performing abduction.
>
> In our opinion the discussion above is primarily regarding semantics and language and not a “severe technical issue” as noted in the reviewers summary.
>
> The reviewer also notes that it is “unclear what is the benefit of probing causality in a pre-trained model which is not designed to capture causality”. To respond to this we first note that **model interpretability has established itself as a fundamental avenue of research** within the deep learning and machine learning communities. This is evident from the large number of related publications (e.g., [5, 7]), as well as associated workshows & tutorials at premier machine learning conferences (e.g., [7,8, 9]). As such, it should be clear that understanding implicit structures in deep generative models is a well motivated research endeavour.
>
> Response part 1/2

---

> > ### Author Response · Authors · 2021-11-30
> > **Re: Post-rebuttal review part 2/2**
> >
> > Furthermore, it is important to note that simply because a model is not designed to infer causal structure does not mean it will not implicitly encode causal structure present in the data. A clear example of this is in the work of [10], who fit linear ICA models to observational data and study the properties of the estimated unmixing matrix (as explained in their work, if the unmixing matrix can be permuted to be lower triangular, then there is evidence of causal structure in the data). We note that linear ICA is a generative model that, as noted by the review, is not “designed to capture causality”; in fact the generative model within linear ICA only assumes independent, non-Gaussian latent variables (a special case of which turns out to be linear, non Gaussian, acyclic causal models). However, the authors in [6] demonstrate that such models can indeed be employed to infer causal structure.
> >
> >
> > [1] Khemakhem, Ilyes, et al. "Causal autoregressive flows." International Conference on Artificial Intelligence and Statistics. PMLR, 2021.
> >
> > [2] Pawlowski, Nick, Daniel C. Castro, and Ben Glocker. "Deep structural causal models for tractable counterfactual inference." arXiv preprint arXiv:2006.06485 (2020).
> >
> > [3] Yoon, Jinsung, James Jordon, and Mihaela Van Der Schaar. "GANITE: Estimation of individualized treatment effects using generative adversarial nets." International Conference on Learning Representations. 2018.
> >
> > [4] Johansson, Fredrik, Uri Shalit, and David Sontag. "Learning representations for counterfactual inference." International conference on machine learning. PMLR, 2016.
> >
> > [5] Ribeiro, Marco Tulio, Sameer Singh, and Carlos Guestrin. "" Why should I trust you?" Explaining the predictions of any classifier." Proceedings of the 22nd ACM SIGKDD international conference on knowledge discovery and data mining. 2016.
> >
> > [6] Lundberg, Scott M., and Su-In Lee. "A unified approach to interpreting model predictions." Proceedings of the 31st international conference on neural information processing systems. 2017.
> >
> > [7] Interpretable Machine Learning Tutoria, CVPR, 2021, https://interpretablevision.github.io/
> >
> > [8] Responsible AI Workshop, ICLR 2021, https://sites.google.com/view/rai-workshop/home
> >
> > [9] EMNLP tutorial on interpretability, https://github.com/Eric-Wallace/interpretability-tutorial-emnlp2020
> >
> > [10] Shimizu, Shohei, et al. "A linear non-Gaussian acyclic model for causal discovery." Journal of Machine Learning Research 7.10 (2006).
> >
> > part 2/2

---

### Official Review · Reviewer_Ak2p · 2021-11-01

**Correctness:** 4
**Technical Novelty And Significance:** 3
**Empirical Novelty And Significance:** 3
**Recommendation:** 6
**Confidence:** 4

**Main Review:**

A strength of this paper is that it clearly presents the methodological ideas and formulations.  If you have a background in causal inference, the approach is relatively straightforward to apply in the latent space, as the methodology is well-laid out in Section 3.  The visualization in Figure 2 is quite nice to illustrate the idea and the utility of the linear classifier.  There are many results and visualizations, allowing the reader to do a great deal of visual evaluation.  The ideas are highly relevant to ethical considerations in modern applications of deep learning (although I would like to see this expanded further in the discussion!).

Where this becomes tricky is on the interpretation of the results. 4.1 and 4.2 are relatively straightforward to evaluate, but on a small and simplistic scale. It is very difficult to evaluate whether the results from this model are good or not on the real data (4.3), which is a common problem in the evaluation of causal models.  In particular, the different results between the different models are very challenging to evaluate, and they need to be discussed at length.  Why do the 3 baselines and the proposed method differ at times?  They are all analyzing the same data, and the "prior" relationship is not necessarily true for the generative model (as is noted by the authors). The authors should discuss at greater length their rationale for the proposed causal relationship, and whether that appears true in the system.  Additionally, I would like greater discussion of the visual samples, which don't always match what I would expect from the system.  For example, adding a mustache to a female example without changing the gender seems like a strange example, as essentially no similar samples exist in the training database.

Many of these causal relationships are evaluated with semi-synthetic data setups, as was done in 4.1 and 4.2.  It does seem like you selectively choose the training images in Section 4.3 to imply specific causal relationships and see if you method picks them up in the latent space, rather than subjectively choosing priors, which may or may not be true in the generative latent space.

The results presented in Table 2 are strange.  Notably, the $G\rightarrow H$ strategy is significantly better on test but significantly worse on validation than $H\rightarrow G$.  Any idea why there is this discrepancy?  Given these results, it is unclear how strong the evidence is for the claim that using the causal direction is better than the anticausal direction or if it is just statistical fluctuation.



**Summary Of The Paper:**

This manuscript proposes CAGE, a method for determining causal relationships between attributes based on the learned representations of deep causal models.  The method is based upon relatively standard causal inference methods after structuring the deep latent representations between them.  The method is straightforward to apply and estimate the Generative Average Treatment Effect (GATE).  Results are shown on both synthetic and real datasets, and are used to infer relationships.

**Summary Of The Review:**

Interesting and potentially useful method in a relevant topic area.  The evaluation approach could be improved.

---

> ### Author Response · Authors · 2021-11-17
> **Response to Reviewer Ak2p Part 1/2**
>
> We thank the reviewer for their detailed comments and helpful feedback on the draft of our paper. We are heartened to hear that the reviewer felt the presentation of our methodology including the main ideas and formulations was clear. We also appreciate that the reviewer felt that the methodology is well laid out in section 3 and that Fig 2 in particular was a useful visual aid in highlighting the utility of the linear classifier. We now respond to the main questions asked by the reviewer below.
>
> **Difficulty in Evaluation**
> The reviewer makes an astute observation in that reliably evaluating causal models is a challenging problem. However, we believe that our results in 4.2 are not quite small and simplistic in scale. Specifically, we considered MorphoMNIST---a dataset that has been used for deep structural equation models and counterfactual inference [1]---and CelebaHQ which are high dimensional image datasets and understanding learned causal structure for generative models in these domains has until this work an unexplored problem. In particular, in the context of MorphoMNIST we are able to control the generative mechanism and thereby verify CAGE gives the correct description of the underlying causal process.
>
> As the reviewer correctly points out in our experiments we find that our three chosen causal discovery baselines are not always consistent. This is often the case with in of causal discovery and arises as a result of different underlying assumptions of each of the methods (for example, ANM accommodates nonlinear causal relationships but stipulates that noise must be additive, in contrast both LiNGAM and NO-TEARS assume linear causal relationships). As such, despite analyzing the same data, it is often the case that distinct causal discovery algorithms obtain different results. An important example of this is [2], where a series of bivariate causal discovery methods were compared on both synthetic and real datasets and the authors report significant differences in the overall performance of various causal discovery algorithms (for example see Figures 11 and 13). In the context of this, it is important to highlight that one key methodological difference between CAGE and other causal discovery methods we study is that CAGE explicitly leverages counterfactual class probabilities when calculating $\delta \tau$. Moreover, in Appendix C we consider an additional ablation experiment where both hard and soft labels are used for the baselines and a separate experiment where counterfactual data is also provided.
>
>
> **Expanded discussions**
>
> ***Ethical implications:*** We appreciate the reviewer noting the importance of this work for modern deep learning applications! Following up on the reviewer’s feedback, we have also expanded on the ethics statement in the revised version of the paper.
>
> ***Visual Samples:*** We acknowledge the reviewers' concern that generating images of females with mustaches without changing the gender might initially be awkward given that the dataset itself has no such examples. However, we argue that under our biological prior the true causal process tells us that gender causes the presence of a mustache and our goal is to then identify if the generative model has learned this causal association or the reverse (or even independence). To do so, our CAGE framework provides a manipulation strategy of latent vectors which can be pushed through the generative model to obtain image samples. As this is a deterministic mapping, we can hope to visually inspect a causal signature on the secondary attribute---e.g. changing gender from Female->Male may introduce a mustache but removing mustaches from males is less likely to change gender to female. Similarly, if the two attributes are independent like Gender and Smiling our visual inspection should inform us that both Males and Females (thankfully!) are capable of smiling which is what the generative model learns in Table 1. We understand that the visual nuance for all attribute pairs was not immediately clear from Table 1 and we have added qualitative discussion in the appendix for all additional generated samples.
>
> Response Part 1/2

---

> > ### Author Response · Authors · 2021-11-17
> > **Response to Reviewer Ak2p Part 2/2**
> >
> > **Clarification on 4.3**
> > We recognize the reviewers comment regarding our experimental protocol in section 4.3. We believe that there might be a small misunderstanding of our experimental protocol which might have conveyed that we selectively choose our training images to imply a causal direction. We would like to push back against this assertion and mention that the original group DRO paper [3] considered the worst group test accuracy of Blonde Males as a challenging task as they are severely underrepresented in CelebA. As a result we perform causal data augmentation using ground truth labels in the dataset of non-blonde males whose hair color is manipulated to blonde and conversely blonde females whose gender is manipulated to being male. In each case, care was taken such that roughly ~10k additional data points were generated for training with no leakage to the test set. We believe this is a fair experimental setup which does not advantage CAGE in any specific way but allows it to amplify the learned causal relationship between gender and blonde hair in the generative model. We also note that our findings using causal data augmentation are in line with similar approaches within the NLP community where there is a nascent appreciation that the causal direction of the data collection process bears nontrivial implications in the domains of Self-Supervised Learning and Domain adaptation [4].
> >
> > With regards to the results in table 2, the reviewer comments that it is unclear whether there is evidence to suggest that “using the causal direction is better than the anti causal direction”. We agree that data augmentation strategies can often succeed or fail for a variety of reasons. As a result, we have added an additional experiment to quantitatively compare the quality of generated images under each causal direction (G->H and H->G) using FID scores, a widely adopted metric used to assess the quality of generated samples. We find that when samples are generated using the causal model (G->H) there is a 19.5% drop in FID scores as compared to samples generated using the anti causal model (H->G). Taken together with the data augmentation results, this provides compelling evidence of the need to understand implicit causal structure present in G.
> >
> > [1] Pawlowski, Nick, Daniel C. Castro, and Ben Glocker. "Deep structural causal models for tractable counterfactual inference." Proceedings of 34th Int. Conf. Neural Information Processing Systems (2020)
> >
> > [2] Mooij, Joris M., et al. "Distinguishing cause from effect using observational data: methods and benchmarks." The Journal of Machine Learning Research 17.1 (2016): 1103-1204.
> >
> > [3] Sagawa, Shiori, et al. "Distributionally robust neural networks for group shifts: On the importance of regularization for worst-case generalization." arXiv preprint arXiv:1911.08731 (2019).
> >
> > [4] Jin, Zhijing, et al. "Causal Direction of Data Collection Matters: Implications of Causal and Anticausal Learning for NLP." arXiv preprint arXiv:2110.03618 (2021).
> >
> > Response Part 2/2

---

> ### Author Response · Authors · 2021-11-23
> **Kind Reminder to Respond to our Rebuttal**
>
> Dear Reviewer,
>
> We thank you for the time and effort spent in reviewing our paper. We wish to inform you that we have updated our manuscript as well as responded to your main concerns in our rebuttal below. We hope our responses were satisfactory in clarifying any remaining points that hinder you from potentially reconsidering your evaluation of our work. We are also happy to respond to any new guidance or questions that you may have.

---

### Official Review · Reviewer_AcYm · 2021-11-02

**Correctness:** 3
**Technical Novelty And Significance:** 4
**Empirical Novelty And Significance:** 4
**Recommendation:** 6
**Confidence:** 4

**Main Review:**

Pros: The paper is well-structured. The idea is novel. The authors established a new method for both counterfactual manipulation and treatment evaluation. The paper also designs corresponding experiments and evaluation methods. The generated counterfacual images and the causal-effect relationship seem reasonable.

Cons: Since the hyperplane that encodes the classification boundary is obtained by linear classifier, it means that the learned attributes should be linearly separable in the latent space. How to guarantee or to proof this, like some theoretical analysis or qualitative feature visualization?

Other comments:
1)  The CAGE performs the binary treatment, i.e., manipulating attributes with binary labels. Is it possible to extend to other attributes on multi-category situations or even continuous situations?
2)  How to obtain the normal vector to the hyperplane given the high-dimensional latent encoding z?
3)  The text of histograms in Table 1 is not clear enough, please check.
4)  Some typos, please check.


**Summary Of The Paper:**

This paper proposes a novel framework CAGE (causal probing of deep generative models) for inferring the causal-effect relationship in deep generative models. The treatment is implemented by moving linearly along the hyperplane normal. Then the attribute treatment effect can be quantified via a proxy classifier. The authors evaluated the CAGE on low-dimensional synthetic and high-dimensional datasets. Results show the ability to infer causal relationships of the proposed CAGE.


**Summary Of The Review:**

This paper is novel on the interpretability by diving into the deep generative models.

---

> ### Author Response · Authors · 2021-11-17
> **Response to Reviewer AcYm Part 1/2**
>
> We thank the reviewer for their thoughtful comments and feedback and we appreciate the fact that the reviewer felt our paper was “well-structured” and the idea as being “novel”. We now answer the main questions and concerns raised by the reviewer grouped by theme.
>
> **Linear Separability of the semantic attributes:**
> The reviewer raises a great point that learning a hyperplane using a linear classifier means that the latent space must---to a certain---degree be linearly separable. While theoretically hard, there exists strong empirical evidence in the literature that for powerful model classes like GANs and normalizing flows (e.g. Glow [1]) this assumption is quite reasonable in practice. For example, Denton et. al [2] empirically find that GAN latent spaces can be linearly separated for semantic attributes found within facial datasets. Moreover, numerous GAN papers propose various conditional generation mechanisms via interpolating using learned hyperplanes for semantically meaningful attributes [3, 4, 5]. However, unlike our work these approaches are not couched within the potential outcomes framework and as a result cannot be used to diagnose causal relationships between attributes of interest. Finally, to alleviate the reviewers concern we also report the test set accuracies of the linear classifiers for all semantic attributes found in section 4.3 Table 1 where we see for most attributes (e.g. Gender) the classification accuracy of the linear latent classifier is quite reasonable.
>
> | Attributes  	| Test Accuracy	|
> |-------------	|---------------------------	|
> | Gender      	| 90.0                      	|
> | Mustache    	| 79.7                      	|
> |Young   	| 75.1                      	|
> | Bald        	| 70.8                      	|
> | Goatee      	| 83.2                      	|
> | Rosy Cheeks | 72.8                      	|
> | Pointy Nose 	| 64.6                      	|
> | Brown Hair  	| 65.4                      	|
> | Smiling     	| 79.6                      	|
> | Blonde     	| 84.8                      	|
>
> Finally, it is important to note that while we have chosen to parameterize counterfactual manipulations as linear projections over the latent space, our framework can absolutely be used in conjunction with any other definition for counterfactual manipulations. As mentioned above, we chose this parameterization due to its widespread use, however, other approaches can also be accomodated.
>
>
> **Extending CAGE to multi-category and continuous attributes:**
> In its current instantiation, CAGE represents the first approach to perform causal discovery for pairs of binary attributes over modern deep generative models operating on high dimensional image data. We agree with the reviewer, as well as explicitly noted in Section 5, that extending CAGE to incorporate multi-category and continuous attributes is a ripe direction for future work.
>
> Here, we outline our thoughts on extending CAGE for such settings. For multi-category attributes, we may learn multiple hyperplanes by posing the problem as a 1 vs. all classification task which would enable counterfactual generations simply by choosing the correct pair of hyperplanes out of all learned hyperplanes for counterfactual generation and estimating treatment effects. For continuous attributes, we need a smooth characterization of the treatment values which could potentially be done by learning a regression fit for the treatment variable and manipulating the latent vectors along this curve for counterfactual generation. While our current formulation has been developed for binary attributes, we find some preliminary qualitative evidence that such a smooth interpolation is indeed plausible in the interpolations in Figure 1 for different alpha.
>
> **Obtaining the normal vector to the hyperplane:**
> The normal vector $h$ to a hyperplane $w$ must satisfy the following equality: $h^Tw = 0$. Thus, we can always find $h$ by solving the given equation for a specific hyperplane $w$.
>
> Response part 1/2

---

> > ### Author Response · Authors · 2021-11-17
> > **Response to Reviewer AcYm Part 2/2**
> >
> > **Presentation Issues:**
> > We thank the reviewer for pointing out the minor presentation issues within our figures and general typos throughout the paper. We plan to fix these in the updated version of our paper.
> >
> > We would like to thank the reviewer for their review of our paper. We believe we have answered all the great points raised by the reviewer in our author response. We respectfully ask the reviewer to reconsider their impression of the paper and potentially improve the given score if the raised concerns have been allayed. We thank the reviewer again for their time and we are also happy to answer any further questions that arise!
> >
> >
> > [1] Kingma, Diederik P., and Prafulla Dhariwal. "Glow: Generative flow with invertible 1x1 convolutions." arXiv preprint arXiv:1807.03039 (2018).
> >
> > [2] Denton, Emily, et al. "Image counterfactual sensitivity analysis for detecting unintended bias." arXiv preprint arXiv:1906.06439 (2019).
> >
> > [3] Ramaswamy, Vikram V., Sunnie SY Kim, and Olga Russakovsky. "Fair attribute classification through latent space de-biasing." Proceedings of the IEEE/CVF Conference on Computer Vision and Pattern Recognition. 2021.
> >
> > [4] Guha Balakrishnan, Yuanjun Xiong, Wei Xia, and Pietro Perona. Towards causal benchmarking of bias in face analysis algorithms. In Proceedings of European Conference on Computer Vision (ECCV), 2020
> >
> > [5] Yujun Shen, Jinjin Gu, Xiaoou Tang, and Bolei Zhou. Interpreting the latent space of GANs for semantic face editing. In Proceedings of the IEEE/CVF Conference on Computer Vision and Pattern Recognition, pages 9243–9252, 2020.
> >
> > Response Part 2/2

---

> ### Author Response · Authors · 2021-11-23
> **Kind Reminder to Respond to our Rebuttal**
>
> Dear Reviewer,
>
> We appreciate the time and energy you've dedicated to your review. We have updated our manuscript and provided a detailed response to the key points raised in our rebuttal below. We would greatly appreciate a response from you and in particular whether our rebuttal has successfully addressed all of your concerns and potentially allowing you to update your score. We are also equally happy to answer any further remaining questions that arise that may help make your evaluation of our work clearer.

---

### Author Response · Authors · 2021-11-23
**Summary of our responses to the reviewers and main updates to manuscript**

We would like to thank all reviewers for their time and valuable feedback when reviewing our paper. We appreciate their constructive criticisms that aided us in updating our draft and thus the overall quality of our paper. We now summarize the main changes to the paper as well as key clarification points used in our rebuttal.

**Linear Separability of Semantic Attributes Reviewer AcYm**

We have empirically addressed our claim that the latent space for generative models is linearly separable for the results reported in Table 1. Specifically, in our rebuttal we reported the test accuracies of the learned linear latent classifiers where we find that for most pairs of attributes the data in the latent space shows a high degree of linear separability. Moreover, we have also highlighted that while we have chosen to parameterize counterfactual manipulations as linear projections over the latent space, our framework can absolutely be used in conjunction with any other definition for counterfactual manipulations. As mentioned above, we chose this parameterization due to its widespread use, however, other approaches can also be accomodated.


**Challenges in Evaluation and description of visual samples Reviewer Ak2p**

We are grateful for the reviewer acknowledging that evaluating causal discovery methods is a challenging task. In our rebuttal we focused on first highlighting how CAGE discovers the correct generative mechanism on MorphoMNIST---a dataset custom made for the study causality in images and deep networks---and yields important insights on the relationship between meta-data (image attributes) in a high dimensional image dataset like CelebAHQ. We also outlined why other causal discovery baselines disagree in their reported results; we believe the principle reason being that each makes a different set of assumptions. With regards to the Reviewer’s comment on ethical implications of our work we have included an expanded discussion in our Ethics statement which can be found in the main draft. Finally, as per the reviewers suggestion we have also added a qualitative description in Appendix D to all additional generated samples for each pair of attributes in Table 1, where we highlighted the key visual markers which may further help interpret the results found by CAGE.

**Motivation and setup Reviewer TgM5**

We acknowledge the reviewers' healthy skepticism over our CAGE framework, but we believe much of this stems from the fact that our problem setup is different from classic studies in causality. With the increased prevalence of deep generative models in machine learning applications, we believe it is crucial to interpret what the model has learned before it can be deployed in high stake settings. The language of causality is a natural vehicle for this direction and using it we seek to understand ***what the generative model*** has captured which is a fundamentally different question to uncovering the (causal) data generative process for the training dataset. This question is also timely and important given the growing use-cases of the generated data, as acknowledged by other reviewers and even more so, in cases where the model’s generative process is different from the true causal structure within the training data. Our work is the first study in this direction, and as such, lays the foundation for future theoretical and empirical research. We hope this and our extended discussion in the rebuttal reconciles any lingering doubt the reviewer might have had regarding the thrust of our work.


**Correlation vs. Causation Reviewer xP5d**

To alleviate the reviewers concern we have added an additional new Appendix E which computes the MMD distance between the counterfactual distribution and the conditional distribution as a function of the interpolant strength. We believe this provides clear evidence that these distributions are not the same and what CAGE seeks to uncover is not any statistical inference query but rather a causal one. To further stress this point we also provided in our rebuttal an example of a situation where there is a clear correlation between attributes---Gender and Smiling---yet CAGE infers that these are actually independent with regards to the trained generative model. We hope that these additional results clarify that our method goes beyond simple statistical correlation and fully addresses the reviewers concerns.

We hope that these explanations and resulting updates to the paper alleviates the main points raised by the reviewers and alsol allow them to endorse the paper more wholeheartedly.

---

### Decision · Program_Chairs · 2022-01-20

**Decision:**

Reject

**Comment:**

This paper proposes the framework CAGE (causal probing of deep generative models) for estimating counterfactuals and unit-level causal effects in deep generative models. CAGE employs geometrical manipulations within the latent space of a generative model to estimate the counterfactual quantities. The estimator is written in potential outcome language and assumes unconfoundedness, positivity, stable unit treatment value assumption (SUTVA), and linear separability in semantic attributes of the latent space. Furthermore, the framework considers only the case of binary treatments.

One major concern raised by reviewers TgM5 and xP5d is that the method is based on a trained generative model, which may not be the true data-generating model. In this case, the paper appears to address statistical dependencies instead of the actual causal relationships in the real world. The authors claim to empirically show that their framework can probe unit-level (individual) causal effects. However, the reviewers are concerned that no theoretical support for the correctness of the method is provided. In other words, the problem is assumed away once a probabilistic model is assumed to be equal to the true generative model, which is almost never the case in practice and is well-known in the field. We want to encourage the authors to provide a more detailed theoretical justification, perhaps with proofs and/or references, that the proposed method can infer causal and counterfactual relationships given the underlying assumptions.

After all, reviewers were interested but somewhat skeptical about the method's ability to learn causal and counterfactual relationships. Unfortunately, the paper is not ready for publication yet. Still, we would like to encourage the authors to take the reviews seriously and try to improve the manuscript accordingly.